 

# eIF2B conformation and assembly state regulate the integrated stress response

**Michael Schoof[1,2], Morgane Boone[1,2†], Lan Wang[1,2†], Rosalie Lawrence[1,2†], Adam Frost[2,3]\*, Peter Walter[1,2]\***

[1]Howard Hughes Medical Institute, University of California at San Francisco, San Francisco, United States; [2]Department of Biochemistry and Biophysics, University of California at San Francisco, San Francisco, United States; [3]Chan Zuckerberg Biohub, San Francisco, United States

**Abstract** The integrated stress response (ISR) is activated by phosphorylation of the translation initiation factor eIF2 in response to various stress conditions. Phosphorylated eIF2 (eIF2-P) inhibits eIF2's nucleotide exchange factor eIF2B, a twofold symmetric heterodecamer assembled from subcomplexes. Here, we monitor and manipulate eIF2B assembly in vitro and in vivo. In the absence of eIF2B's $\alpha$-subunit, the ISR is induced because unassembled eIF2B tetramer subcomplexes accumulate in cells. Upon addition of the small-molecule ISR inhibitor ISRIB, eIF2B tetramers assemble into active octamers. Surprisingly, ISRIB inhibits the ISR even in the context of fully assembled eIF2B decamers, revealing allosteric communication between the physically distant eIF2, eIF2-P, and ISRIB binding sites. Cryo-electron microscopy structures suggest a rocking motion in eIF2B that couples these binding sites. eIF2-P binding converts eIF2B decamers into 'conjoined tetramers' with diminished substrate binding and enzymatic activity. Canonical eIF2-P-driven ISR activation thus arises due to this change in eIF2B's conformational state.

**\*For correspondence:**
adam.frost@ucsf.edu (AF);
peter@walterlab.ucsf.edu (PW)

[†]These authors contributed equally to this work

## Introduction

All cells must cope with stress, ranging from nutrient deprivation to viral infection to protein misfolding. Cell stress may arise from cell-intrinsic, organismal, or environmental insults, yet often converges on common regulatory nodes. The integrated stress response (ISR) is a conserved eukaryotic stress response that senses and integrates diverse stressors and responds by reprogramming translation (*Harding et al., 2003*). ISR activation has been linked to numerous human diseases, including cancer and neurological diseases (reviewed in *Costa-Mattioli and Walter, 2020*). While acute ISR activation largely plays a cytoprotective role, its dysregulation (both aberrant activation and insufficient activation) can negatively affect disease progression. In many pathological conditions, for example, the ISR is constitutively activated and maladaptive effects arise that worsen the disease outcome. Many conditions of cognitive dysfunction, for example, have been linked causally to ISR activation in mouse models, including brain trauma resulting from physical brain injuries (*Chou et al., 2017*; *Sen et al., 2017*), familial conditions including Vanishing White Matter Disease and Down syndrome (*Leegwater et al., 2001*; *van der Knaap et al., 2002*; *Zhu et al., 2019*), neurodegenerative diseases such as Alzheimer's and amyotrophic lateral sclerosis (ALS) (*Atkin et al., 2008*; *Ma et al., 2013*), and even the cognitive decline associated with normal aging (*Sharma et al., 2018*; *Krukowski et al., 2020*). Our understanding of the molecular mechanism of ISR regulation therefore is of profound importance.

Translation reprogramming upon ISR induction results as a consequence of reduced ternary complex (TC) levels. The TC is composed of methionyl initiator tRNA (Met-tRNA$^i$), the general translation initiation factor eIF2, and GTP (*Algire et al., 2005*). At normal, saturating TC concentrations, translation initiates efficiently on most mRNAs containing AUG translation start sites; however, translation

of some mRNAs is inhibited under these conditions by the presence of inhibitory small upstream open reading frames (uORF) in their 5′ untranslated regions (UTRs; *Hinnebusch et al., 2016*). When TC levels are sub-saturating, translation is repressed on most mRNAs. In contrast, some mRNAs that contain uORFs in their 5′ UTRs are now preferentially translated, including mRNAs encoding stress-responsive transcription factors, such as ATF4 (*Harding et al., 2000*). Thus, TC availability emerges as a prime factor in determining the translational and, consequentially, transcriptional programs of the cell.

The central mechanism that regulates TC levels in response to stress conditions concerns the loading of eIF2's γ subunit with GTP. Without GTP, eIF2 cannot bind Met-tRNA$^i$ and hence does not assemble the TC. Loading is catalyzed by the guanine nucleotide exchange factor (GEF) eIF2B, a large decameric and twofold symmetric enzyme that is composed of two copies each of five differ-ent subunits, eIF2Bα, β, δ, γ, and ε (*Kashiwagi et al., 2016*; *Tsai et al., 2018*; *Wortham et al., 2014*; *Zyryanova et al., 2018*). Stress sensing is accomplished by four upstream kinases (PKR, PERK, GCN2, and HRI) that are activated by different stress conditions and, in turn, phosphorylate eIF2 as a common target (*Hinnebusch, 2005*; *Guo et al., 2020*; *Dey et al., 2005*; *Shi et al., 1998*). Phos-phorylation by each of these kinases converges on a single amino acid, S51, in eIF2's α subunit (eIF2α). As a profound consequence of eIF2α S51 phosphorylation, eIF2 converts from eIF2B's sub-strate for GTP exchange into a potent eIF2B inhibitor.

Cryo-electron microscopy (cryo-EM) studies of eIF2B•eIF2 complexes show that eIF2 snakes across the surface of eIF2B in an elongated conformation, contacting eIF2B at four discontinuous interfaces, which we here refer to as IF1–IF4 (*Figure 1—figure supplement 1*; *Kenner et al., 2019*; *Gordiyenko et al., 2019*; *Kashiwagi et al., 2019*; *Adomavicius et al., 2019*). IF1 and IF2 engage eIF2γ (containing eIF2's GTPase domain) with eIF2Bε, sandwiching eIF2γ between eIF2Bε's catalytic and core-domain, respectively. This interaction pries the GTP binding site open, thus stabilizing the apostate to catalyze nucleotide exchange. IF3 and IF4 engage eIF2 via its α subunit across eIF2B's twofold symmetry interface, where two eIF2Bβδγε tetramer subcomplexes are joined. The eIF2α binding surfaces line a cleft between eIF2Bβ (IF3) and eIF2Bδ′ (IF4) (the prime to indicate the subunit in the adjoining tetramer). Upon S51 phosphorylation, eIF2α adopts a new conformation that ren-ders it incompatible with IF3/IF4 binding (*Bogorad et al., 2017*; *Kenner et al., 2019*; *Zhu et al., 2019*; *Kashiwagi et al., 2019*; *Adomavicius et al., 2019*; *Gordiyenko et al., 2019*). Rather, phos-phorylation unlocks an entirely new binding mode on the opposite side of eIF2B, where eIF2α-P now binds to a site between eIF2Bα and eIF2Bδ. We and others previously proposed that, when bound to eIF2B in this way, the β and especially the γ subunits of eIF2-P could sterically block eIF2γ of a concomitantly bound unphosphorylated eIF2 substrate from engaging productively with eIF2Bε's active site (*Kashiwagi et al., 2019*; *Kenner et al., 2019*). Such a blockade could explain the inhibitory effect of eIF2-P, and this model predicts that GEF inhibition should depend on eIF2γ as the entity responsible for causing the proposed steric clash.

Both eIF2 and eIF2-P binding sites span interfaces between eIF2B subunits present in the deca-mer but not in the subcomplexes from which it is assembled. The eIF2B decamer is built from two eIF2Bβδγε tetramers and one eIF2Bα$_2$ homodimer (*Wortham et al., 2014*; *Tsai et al., 2018*). These subcomplexes are stable entities that, when mixed in vitro, readily assemble into decamers. The eIF2Bβδγε tetramer has a low, basal GEF activity as it can only engage with eIF2 through IF1–IF3 (*Tsai et al., 2018*; *Craddock and Proud, 1996*). As expected, eIF2B decamer assembly results in a greater than twentyfold rate enhancement of nucleotide exchange, presumably due to enhanced substrate binding caused by the completion of the eIF2α binding site through the addition of IF4 (*Tsai et al., 2018*; *Craddock and Proud, 1996*). Assembly of the eIF2B decamer is driven by eIF2Bα$_2$, which acts as an assembly-promoting factor. Thus, eIF2B assembly into a decamer allows the modalities of (i) full GEF activity on eIF2 and (ii) inhibition by eIF2-P to manifest.

The activity of the ISR can be attenuated by ISRIB, a potent small drug-like molecule with dra-matic effects (*Sidrauski et al., 2013*). In mice, ISRIB corrects with no overt toxicity the cognitive defi-cits caused by traumatic brain injury (*Chou et al., 2017*), Down syndrome (*Zhu et al., 2019*), normal aging (*Krukowski et al., 2020*), and other brain dysfunctions (*Wong et al., 2018*) with an extraordi-nary efficacy, indicating that the molecule reverses the detrimental effects of a persistent and mal-adaptive state of the ISR. ISRIB also kills metastatic prostate cancer cells (*Nguyen et al., 2018*). ISRIB's mechanistic target is eIF2B to which it binds in a binding groove that centrally bridges the symmetry interface between eIF2Bβδγε tetramers (*Sekine et al., 2015*; *Tsai et al., 2018*;

*Zyryanova et al., 2018*; *Sidrauski et al., 2015*). As such, it acts as a 'molecular staple', promoting assembly of two eIF2Bβδγε tetramers into an enzymatically active eIF2B(βδγε)$_2$ octamer. Here, we further interrogated the role of ISRIB by engineering cells that allow us to monitor and experimentally manipulate eIF2B's assembly state. These experiments led to the discovery of a conformational switch that negatively couples the eIF2 and eIF2-P binding sites and the ISRIB binding site by allosteric communication in the eIF2B complex. This conformational switch is the central mechanism by which ISR activation occurs.

## Results

### eIF2B assembly state modulates the ISR in cells

To investigate the role of eIF2B's assembly state in controlling ISR activation, we developed ISR reporter cells that enable experimental modulation of the eIF2B decamer concentration. To this end, we tagged eIF2Bα with an FKBP12$^{F36V}$ degron in human K562 cells (*Figure 1—figure supplement 2A, B*), using CRISPR-Cas9 to edit the endogenous locus. The cell-permeable small molecule dTag13 induces selective degradation of the FKBP12$^{F36V}$-tagged eIF2Bα (*Figure 1A*; *Nabet et al., 2018*). We also engineered a genomically integrated dual ISR reporter system into these cells. The reporter system consists of the mNeonGreen fluorescent protein placed under translational control of a uORF-containing 5′UTR derived from ATF4 ('ATF4 reporter') and the mScarlet-i fluorescent protein containing a partial ATF4 5′ UTR from which the uORFs have been removed ('general translation reporter'). To optimize the signal of these reporters, we fused both fluorescent proteins to the ecDHFR degron (*Figure 1—figure supplement 3*). This degron drives the constitutive degradation of the fusion proteins unless the small molecule trimethoprim is added to stabilize them (*Iwamoto et al., 2010*). In this way, the reporters allow us to monitor only de novo translation. Unless otherwise stated, trimethoprim was added concurrently with other treatments.

Treating ISR reporter cells with the small molecule dTag13 led to a rapid and complete degradation of FKBP12$^{F36V}$-tagged eIF2Bα (*Figure 1B*). As expected, eIF2Bα degradation was selective, as eIF2Bδ, which binds directly to eIF2Bα in the decamer, remained intact. dTag13 treatment also did not increase eIF2α phosphorylation, a hallmark of canonical ISR activation by ISR kinases (*Figure 1B*). Nevertheless, dTag13-induced eIF2Bα degradation led to increased translation of the ATF4 reporter and decreased translation of the general translation reporter (*Figure 1C*. *Figure 1—figure supplement 4A*) in a concentration-dependent manner. dTag13 treatment of cells lacking FKBP12$^{F36V}$ degron-tagged eIF2Bα did not induce the ISR (*Figure 1—figure supplement 5*). These results demonstrate that ISR-like translational reprogramming follows eIF2Bα depletion.

### ISRIB resolves assembly-based stress

As predicted from previous in vitro work, ISRIB entirely reversed the ISR translational reprogramming by eIF2Bα depletion (EC$_{50}$ = 1.4 nM; *Figure 1D, Figure 1—figure supplement 4B*; *Tsai et al., 2018*). Thus, eIF2Bα can be quantitatively replaced by ISRIB, a small molecule that causes eIF2B (βδγε)$_2$ octamer assembly, rendering the eIF2B decamer and ISRIB-stabilized octamer functional equivalents in these cells. dTag13 treatment led to continued increases in ATF4 translation and decreased general translation over a 6 hr window (*Figure 1E, Figure 1—figure supplement 4C*), and co-treatment with ISRIB completely reversed ISR activation.

By contrast, ISRIB inhibited eIF2-P-based stress induced by thapsigargin treatment only at early timepoints (1–3 hr), whereas at later timepoints, ISRIB showed greatly diminished effects in blocking ISR activation. These data distinguish eIF2B assembly-based stress and eIF2-P-based stress in their response to mitigation by ISRIB.

### FRET reporters monitor eIF2B assembly state

To directly measure eIF2B's assembly state, we tagged eIF2B subunits with fluorescent protein pairs and used Förster resonance energy transfer (FRET) as a readout of their molecular proximity. We tagged the C-terminus of eIF2Bβ with mNeonGreen as the FRET donor and the C-terminus of eIF2Bδ with mScarlet-i as the FRET acceptor. In this arrangement, donor and acceptor proteins would be in the range of 120–140 Å apart in the eIF2Bβδγε tetramer (expected negligible FRET efficiency) and become juxtaposed at a distance closer to 60–80 Å when two eIF2B tetramers assemble

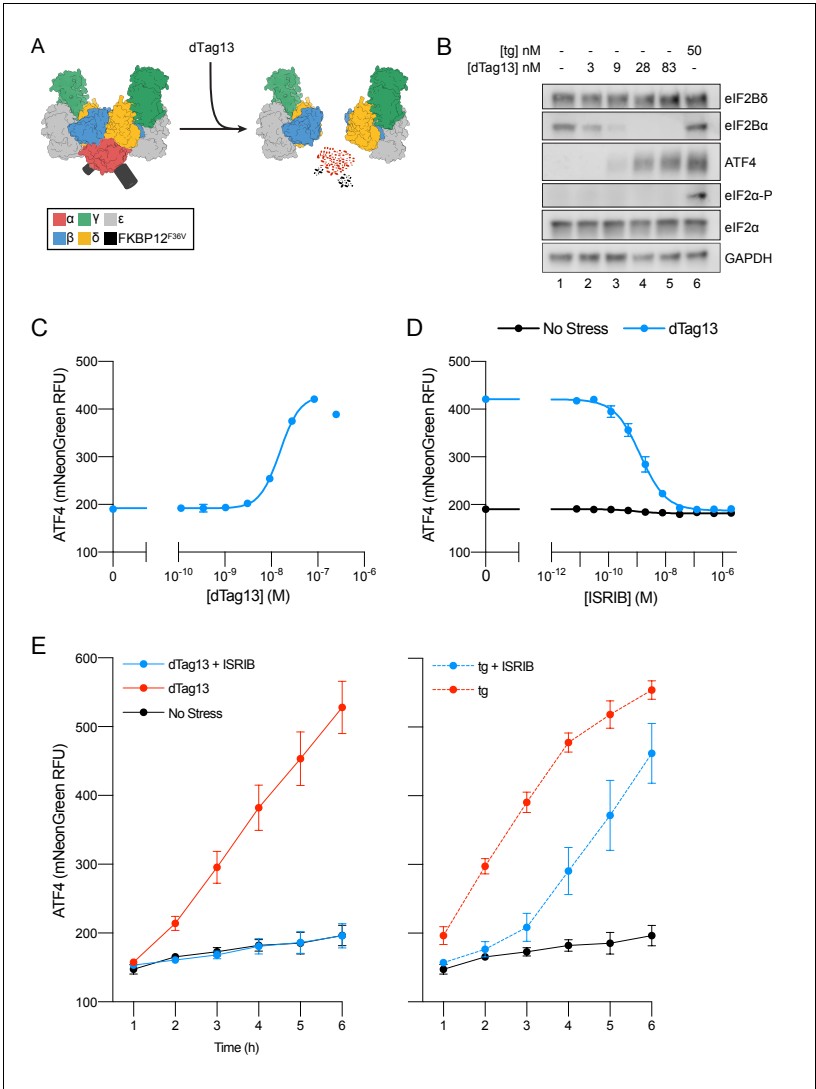

**Figure 1.** Cellular eIF2B assembly state in cells modulates the integrated stress response (ISR). (**A**) Schematic of eIF2B assembly state modulation via the FKBP12$^{F36V}$/dTag13 system used to induce degradation of eIF2Bα. (**B**) Western blot of K562 cell extracts after treatment with thapsigargin (tg) or dTag13 for 3 hr as indicated. Thapsigargin induces the ISR by depleting $Ca^{2+}$ levels in the endoplasmic reticulum. Loading of all lanes was normalized to total protein. (**C–E**) ATF4 reporter levels as monitored by flow cytometry. Trimethoprim was at 20 μM. (**C**) Samples after 3 hr of dTag13 treatment ($EC_{50}$ = 15 nM; s.e.m. = 1 nM). (**D**) Samples after 3 hr of ISRIB treatment ± 83 nM dTag13 ($EC_{50}$ = 1.4 nM; s.e.m. = 0.3 nM). (**E**) Timecourse of tg treatment (dTag13 = 83 nM, tg = 100 nM, ISRIB = 2 μM). For (**B**), eIF2Bδ, eIF2Bα, and GAPDH blots, and the ATF4 and eIF2α blots are from the same gels, respectively; the eIF2α-P blot is from its own gel. For (**C–E**), biological replicates: n = 3. All error bars represent s.e.m.

The online version of this article includes the following figure supplement(s) for figure 1:

**Figure supplement 1.** Overview of key eIF2 and eIF2B interaction surfaces.

**Figure supplement 2.** Tagging of eIF2B subunits in K562 cells.

**Figure supplement 3.** Integrated stress response (ISR) reporter design.

**Figure supplement 4.** Decreases in general translation after eIF2Bα depletion.

**Figure supplement 5.** dTag13 treatment alone does not activate the integrated stress response (ISR).

---

into an octamer or a decamer (expected moderate FRET efficiency). Therefore, this genetically encodable system promised to provide us with a quantitative assay of eIF2B's assembly state.

To first characterize these tools in vitro, we co-expressed the fluorescently tagged eIF2Bβ and eIF2Bδ fusion proteins together with untagged eIF2Bγ and eIF2Bε in *Escherichia coli* and purified the

tetramer as previously described (*Tsai et al., 2018*). Analysis by analytical ultracentrifugation following absorbance at 280 nm demonstrated that the fluorescent protein tags do not interfere with tetramer stability (*Figure 2—figure supplement 1*). Moreover, consistent with our previous work, addition of separately expressed eIF2Bα homodimers (eIF2Bα₂) readily assembled fluorescently tagged eIF2Bβδγε tetramers (eIF2Bβδγε-F) into complete eIF2B decamers. Similarly, the addition of ISRIB caused the tagged tetramers to assemble into octamers.

Upon donor excitation at 470 nm, we next monitored the ratio of fluorescence at 516 nm (donor peak) and 592 nm (acceptor peak) as a function of eIF2Bα₂ and ISRIB concentrations. The results validated our system: in both cases, the FRET signal reliably reported on eIF2Bβδγε-F tetramer assembly into the respective larger complexes with half-maximal assembly ($EC_{50}$) at 250 nM of ISRIB and 20 nM of eIF2Bα₂ (*Figure 2B, C*). Kinetic analysis showed that eIF2Bα₂ drives assembly of eIF2Bβδγε-F tetramers into decamers with a $t_{1/2}$ of 7 min and that ISRIB drives eIF2Bβδγε-F tetramers into octamers with similar kinetics ($t_{1/2}$ = 5 min) (*Figure 2D, E*; 0–55 min time window). By contrast, the dissociation kinetics of eIF2Bα₂-stabilized decamers and ISRIB-stabilized octamers differed substantially. Spiking in an excess of unlabeled eIF2Bβδγε tetramers to trap dissociated eIF2Bβδγε-F tetramers into dark complexes revealed slow eIF2Bα₂-stabilized decamer dissociation kinetics ($t_{1/2}$ = 3 h), whereas ISRIB-stabilized octamers dissociated much faster ($t_{1/2}$ = 15 min) (*Figure 2D, E*; 55–150 min time window).

Still in vitro, as expected, co-treatment of ISRIB and eIF2Bα₂ did not induce greater complex assembly when eIF2Bα₂ was at saturating concentrations (*Figure 2F*). However, ISRIB substantially enhanced complex stability, slowing the dissociation rate of the ISRIB-stabilized decamer such that

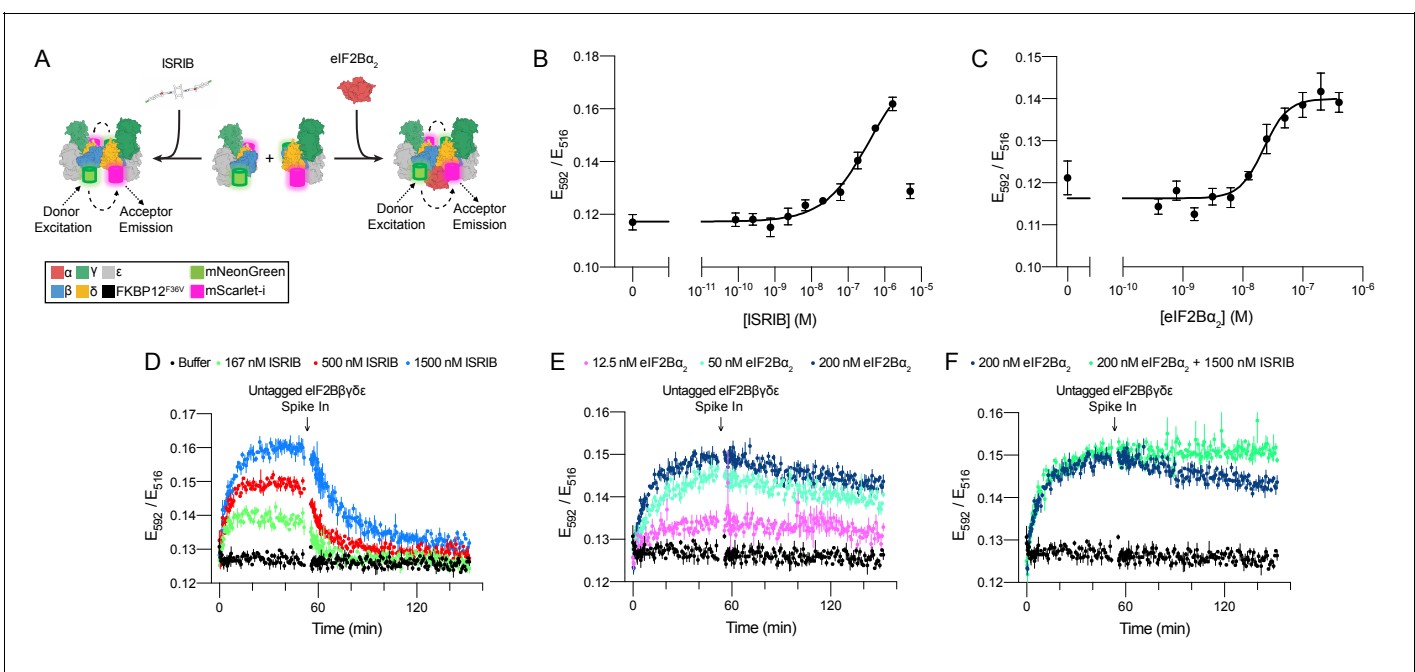

**Figure 2.** Förster resonance energy transfer (FRET) system monitors eIF2B assembly state. (**A**) Schematic depicting the principle of eIF2B assembly state modulation by ISRIB and eIF2Bα₂ and FRET readout. (**B, C**) FRET signal ($E_{592}/E_{516}$) measured after 1 hr of incubation with (**B**) ISRIB ($EC_{50}$ = 250 nM; s.e.m. = 80 nM) or (**C**) eIF2Bα₂ ($EC_{50}$ = 20 nM; s.e.m. = 4 nM) at 50 nM eIF2Bβδγε-F. (**D–F**) Timecourse monitoring of FRET signal ($E_{592}/E_{516}$) after addition of (**D**) ISRIB (association $t_{1/2}$ = 5.1 min, s.e.m. = 0.5 min; dissociation $t_{1/2}$ = 15 min, s.e.m. = 1 min), (**E**) eIF2Bα₂ (association $t_{1/2}$ = 7.3 min, s.e. m. = 0.6 min; dissociation $t_{1/2}$ = 180 min, s.e.m. = 10 min), or (**F**) ISRIB + eIF2Bα₂ (association $t_{1/2}$ = 7 min, s.e.m. = 1 min; dissociation $t_{1/2}$ = N/A) at 50 nM eIF2Bβδγε-F. At t = 52 min, unlabeled eIF2Bβδγε was added to a final concentration of 1 μM. For (**B, C**), representative replicate averaging four technical replicates is shown. For (**D–F**), representative replicate averaging three technical replicates is shown. For (**B–F**), biological replicates: n = 3. All error bars represent s.e.m.

The online version of this article includes the following figure supplement(s) for figure 2:

**Figure supplement 1.** eIF2Bβδγε-F can octamerize and decamerize.

**Figure supplement 2.** Validation of eIF2Bβδγε-F kinetics.

**Figure supplement 3.** ISRIB treatment does not impact guanine nucleotide exchange factor (GEF) activity when eIF2Bα₂ is saturating.

no discernible dissociation was observed. Critically, pre-addition of excess untagged eIF2Bβδγε and tetramer dimerizers (either eIF2Bα$_2$ or ISRIB) led to no change in FRET signal above baseline (*Figure 2—figure supplement 2A–C*). This observation confirms that the lack of signal loss in the ISRIB-stabilized decamer is indeed due to increased complex stability and not to sequestering of dimerizer by the untagged tetramer. Consistent with these observations, treatment with ISRIB at saturating eIF2Bα$_2$ concentrations did not lead to a further increase in eIF2B's nucleotide exchange activity as monitored by BODIPY-FL-GDP nucleotide exchange (*Figure 2—figure supplement 3*).

## eIF2B exists as a decamer in K562 cells

Turning to live cells to monitor and modulate the assembly state of eIF2B, we engineered K562 cells to contain both the FRET reporters (eIF2Bβ-mNeonGreen-FLAG and eIF2Bδ-mScarlet-i-myc) and eIF2Bα-FKBP12$^{F36V}$ (*Figure 1—figure supplement 2A, B*). Consistent with our data on the ISR reporter in *Figure 1*, degradation of eIF2Bα led to translation of ATF4, whereas eIF2α-P and eIF2Bδ levels remain unchanged (*Figure 3A*).

Importantly, degradation of eIF2Bα via dTag13 treatment led to eIF2B complex disassembly, as monitored by FRET signal (*Figure 3B*), validating that our FRET system robustly reports on the eIF2B complex assembly state in living cells. At the 3 hr timepoint, the EC$_{50}$ for eIF2B disassembly was 5 nM (*Figure 3B*), which mirrors the EC$_{50}$ for ISR activation (15 nM, *Figure 1B*). These data indicate that eIF2B's assembly state is intimately linked to translational output.

## ISRIB inhibits the ISR without impacting eIF2B's assembly state

We next treated cells with a titration of ISRIB ± the addition of optimal dTag13 concentration (83 nM, plateau from *Figure 1B*; *Figure 3B*) for 3 hr (*Figure 3C*). ISRIB assembled tetramers into octamers when the eIF2Bα subunit was not present. Notably, in the presence of eIF2Bα, the FRET signal remained unchanged upon increasing ISRIB concentrations, indicating that the assembly state of eIF2B in K562 cells is largely decameric unless eIF2Bα is compromised.

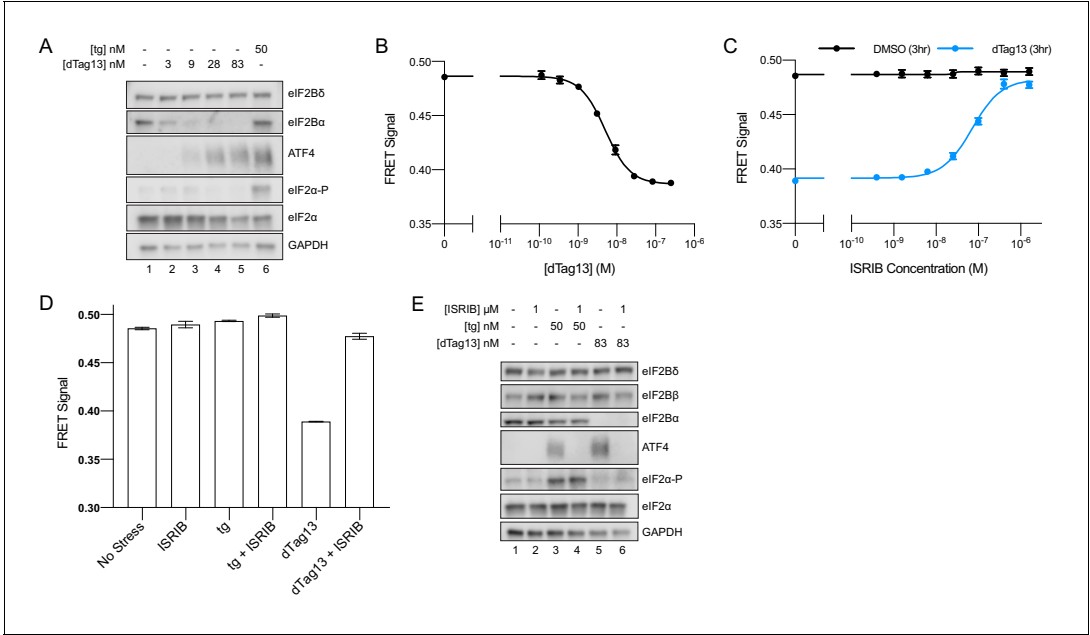

**Figure 3.** eIF2B is a decamer in both unstressed and stressed cells, and ISRIB blocks integrated stress response (ISR) activation. (**A**) Western blot of K562 ISR reporter cell extracts after treatment with thapsigargin (tg) or dTag13 for 3 hr as indicated. (**B–D**) Förster resonance energy transfer signal as monitored by flow cytometry after 3 hr treatment with (**B**) dTag13 (EC$_{50}$ = 5.1 nM; s.e.m. = 0.2 nM), (**C**) ISRIB ± 83 nM dTag13 (EC$_{50}$ = 80 nM; s.e.m. = 10 nM), and (**D**) various stressors (83 nM dTag13, 50 nM tg, ± 1.6 μM ISRIB). The ratio of mScarlet-i/mNeonGreen emission is presented. (**E**) Western blot of K562 ISR reporter cell extracts treated for 3 hr with ISRIB, tg, and/or dTag13 as indicated. All lanes across gels were loaded with equal total protein. For (**A**), eIF2Bδ, eIF2Bα, and GAPDH blots, and the ATF4 and eIF2α blots are from the same gels, respectively; the eIF2α-P blot is from its own gel. For (**E**), eIF2Bδ, eIF2Bβ, and GAPDH blots, ATF4 and eIF2α blots, and eIF2Bα and eIF2α-P blots are from the same gels, respectively. For (**B–D**), biological replicates: n = 3. All error bars represent s.e.m.

As ISRIB's effect on translation is only noticeable upon cellular stress, we wondered whether the assembly state of eIF2B could be affected by stress. To this end, we treated cells with thapsigargin ± ISRIB. We observed no decrease in FRET signal upon ER stress or ISRIB treatment, arguing that eIF2B exists as a fully assembled decamer in both stressed and unstressed cells (*Figure 3D*).

Nevertheless, ISRIB resolved both eIF2-P-based activation of the ISR induced by thapsigargin and assembly-based activation of the ISR induced by eIF2Bα depletion (*Figure 3E*, lanes 4 and 6), implying that while ISRIB does not alter eIF2B's assembly state in the thapsigargin-treated cells, it still impacts ISR signaling. Thus, ISRIB must somehow overcome the inhibition of eIF2B's GEF activity asserted by eIF2-P binding.

## ISRIB blocks eIF2-P binding to eIF2B

To resolve this paradox, we immunoprecipitated eIF2B complexes, pulling on eIF2Bβ-mNeonGreen-FLAG, to assess whether eIF2-P binding changes upon ISRIB treatment in thapsigargin-stressed cells (*Figure 4A*). Consistent with canonical ISR activation, in total cell lysate eIF2α-P levels increased upon stress to a similar extent with and without ISRIB treatment. At the same time, ATF4 translation occurred in stressed cells only, and ISRIB treatment inhibited ATF4 translation (*Figure 4A*, Cell Lysate).

Surprisingly, we found that the amount of eIF2α-P bound to eIF2B was dramatically reduced in the immunoprecipitations from ISRIB-treated cells (*Figure 4A*, eIF2B-Bound). Because the amount of total eIF2α bound by eIF2B is likewise reduced, this result suggests that under these stress conditions the majority of eIF2B-bound eIF2 still associated after immunoprecipitation is phosphorylated (note that the eIF2 antibody used in this analysis detects both eIF2α and eIF2α-P). Thus, ISRIB antagonizes eIF2-P binding to eIF2B. Because the binding sites for ISRIB and eIF2-P are ~50 Å apart, this result suggests an allosteric rather than an orthosteric interplay between ISRIB and eIF2-P binding.

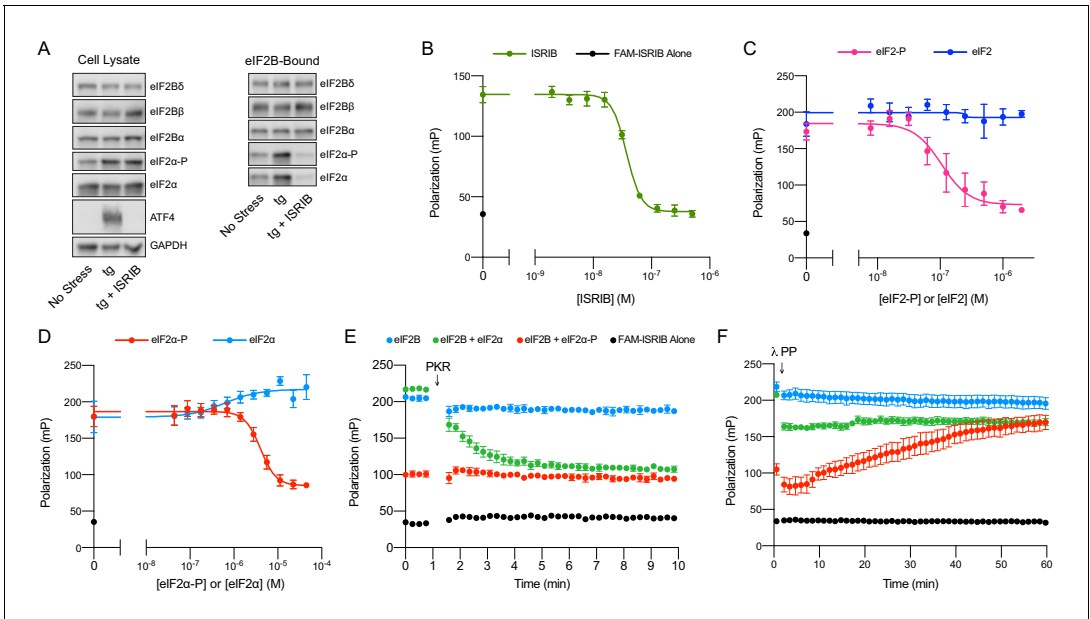

**Figure 4.** ISRIB and eIF2-P compete for eIF2B binding. (A) Western blot of K562 integrated stress response (ISR) reporter cell extracts after treatment with tg ± ISRIB as indicated (left panel) or of eIF2B-bound fraction isolated by anti-FLAG immunoprecipitation of the eIF2B-mNeonGreen-FLAG tagged subunit under native conditions (right panel). (B–D) Plot of fluorescence polarization signal after incubation of FAM-ISRIB (2.5 nM) with 100 nM eIF2B ($\alpha\beta\gamma\delta\epsilon$)$_2$ and varying concentrations of (B) ISRIB (IC$_{50}$ = 37 nM; s.e.m. = 1 nM), (C) eIF2 or eIF2-P (IC$_{50}$ = 210 nM; s.e.m. = 120 nM), and (D) eIF2α or eIF2α-P (IC$_{50}$ = 4000 nM; s.e.m. = 200 nM). (E–F) Timecourse of fluorescence polarization signal after addition of (E) eIF2α kinase PKR and ATP or (F) λ phosphatase. FAM-ISRIB was at 2.5 nM. eIF2B($\alpha\beta\gamma\delta\epsilon$)$_2$ was at 100 nM. eIF2α and eIF2α-P were at 5.6 μM. In (A), eIF2Bδ, eIF2Bα, and eIF2α blots, eIF2Bβ and eIF2α-P blots, and ATF4 and GAPDH blots are from the same gels, respectively. All cell lysate or eIF2B-bound lanes across all gels were loaded with equal total protein. Biological replicates: (B) n = 3; (C) n = 5 (n = 4 at 2 μM); and (D–F) n = 3. All error bars represent s.e.m.

## eIF2α-P is sufficient to impair ISRIB binding to eIF2B

To test this notion, we next examined whether, reciprocally, eIF2-P inhibits ISRIB binding in vitro. To this end, we used a fluorescent ISRIB analog (FAM-ISRIB) that emits light with a higher degree of polarization when bound to eIF2B compared to being free in solution (*Zyryanova et al., 2018*). As previously shown, ISRIB competed with FAM-ISRIB for eIF2B binding (*Figure 4B*; *Zyryanova et al., 2018*). Indeed, our results show that eIF2-P, but not eIF2, competes with FAM-ISRIB binding (*Figure 4C*). In fact, eIF2α-P, that is, eIF2's phosphorylated α-subunit alone, but not eIF2α, its unphosphorylated form, suffices in this assay (*Figure 4D*). This observation defines eIF2α-P as the minimal unit needed to affect ISRIB release.

We confirmed this model with assays that used the eIF2 kinase PKR to phosphorylate eIF2α, thereby over time converting this previously inert component into eIF2α-P, the ISRIB binding antagonist (*Figure 4E*). Conversely, dephosphorylation of eIF2α-P by λ phosphatase over time destroyed its ability to dislodge FAM-ISRIB (*Figure 4F*). Together, these data show that ISRIB binding and eIF2α-P or eIF2-P binding are mutually exclusive events.

## eIF2α-P is sufficient to inhibit eIF2B GEF activity

We further extend these conclusions with activity-based assays. As previously shown, in nucleotide exchange assays that monitor eIF2B's GEF activity towards eIF2, eIF2-P inhibited eIF2B GEF activity in a concentration-dependent manner (*Figure 5A*; *Wong et al., 2018*). ISRIB partially rescued the activity (*Figure 5C*). Remarkably, the phosphorylated α subunit alone (eIF2α-P) inhibited eIF2B GEF activity (*Figure 5B*), and ISRIB again partially rescued activity (*Figure 5D*). This observation is inconsistent with previous models that emphasized the potential for a steric clash between the γ subunit of eIF2-P and the γ subunit of the substrate eIF2 (*Kenner et al., 2019*; *Kashiwagi et al., 2019*).

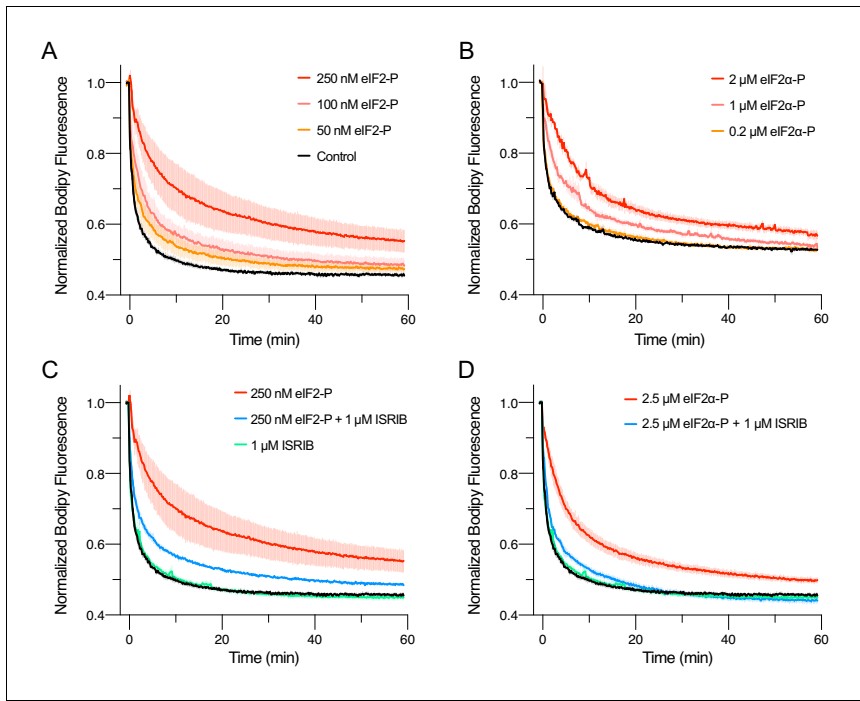

**Figure 5.** eIFα-P is the minimal unit needed to inhibit nucleotide exchange by eIF2B. (A–D) Guanine nucleotide exchange factor (GEF) activity of eIF2B as assessed by BODIPY-FL-GDP exchange. eIF2B(αβδγε)$_2$ was at 10 nM throughout. For (A) $t_{1/2}$ = 1.6 min (Control), 2.5 min (50 nM eIF2-P), 3.5 min (100 nM eIF2-P), and 7.2 min (250 nM eIF2-P). For (B) $t_{1/2}$ = 2.4 min (Control), 3.0 min (0.2 μM eIF2α-P), 5.0 min (1 μM eIF2α-P), and 6.7 min (2 μM eIF2α-P). For (C) $t_{1/2}$ = 1.6 min (Control), 1.9 min (1 μM ISRIB), 3.1 min (250 nM eIF2-P + 1 μM ISRIB), and 7.2 min (250 nM eIF2-P). For (D) $t_{1/2}$ = 1.6 min (Control), 1.9 min (1 μM ISRIB), 3.1 min (2.5 μM eIF2α-P + 1 μM ISRIB), and 5.3 min (2.5 μM eIF2α-P). All error bars represent s.e.m. Biological replicates: (A–D) n = 3 except for the 100 and 50 nM eIF2-P conditions in (A), where n = 2.

Therefore, these data support the notion that the phosphorylated α subunit of eIF2 alone suffices to modulate eIF2B activity, that is, that orthosteric competition cannot wholly explain eIF2-P's inhibitory properties and that the remaining eIF2 subunits are dispensable for this effect.

## eIF2α-P decreases eIF2B's enzymatic activity and antagonizes eIF2 binding

To explain how eIF2α-P alone could block GEF activity, we considered three principal options: (i) eIF2α-P may decrease the rate of eIF2B's enzymatic activity, (ii) allosterically inhibit eIF2 binding to eIF2B, or (iii) perform some combination of those mechanisms. To investigate the relative contributions of these mechanisms, we employed multiple turnover kinetic measurements of eIF2B activity at varying eIF2 concentrations. We measured the initial velocity of this reaction and performed Michaelis–Menten analysis to determine $V_{max}$ and $K_M$ of the GEF reaction at varying concentrations of eIF2α-P (*Figure 6A, Figure 6—figure supplement 1*). Notably, with increasing concentrations of eIF2α-P, $V_{max}$ decreased while $K_M$ increased, suggesting that both substrate affinity and eIF2B catalytic activity were affected by eIF2α-P binding. We next examined how inhibited eIF2B decamers compared to tetramers. Intriguingly, at near-saturating eIF2α-P concentrations, the $k_{cat}/K_M$ ratio, a measure of specific enzyme activity, approached that of the eIF2Bβδγε tetramer, suggesting that

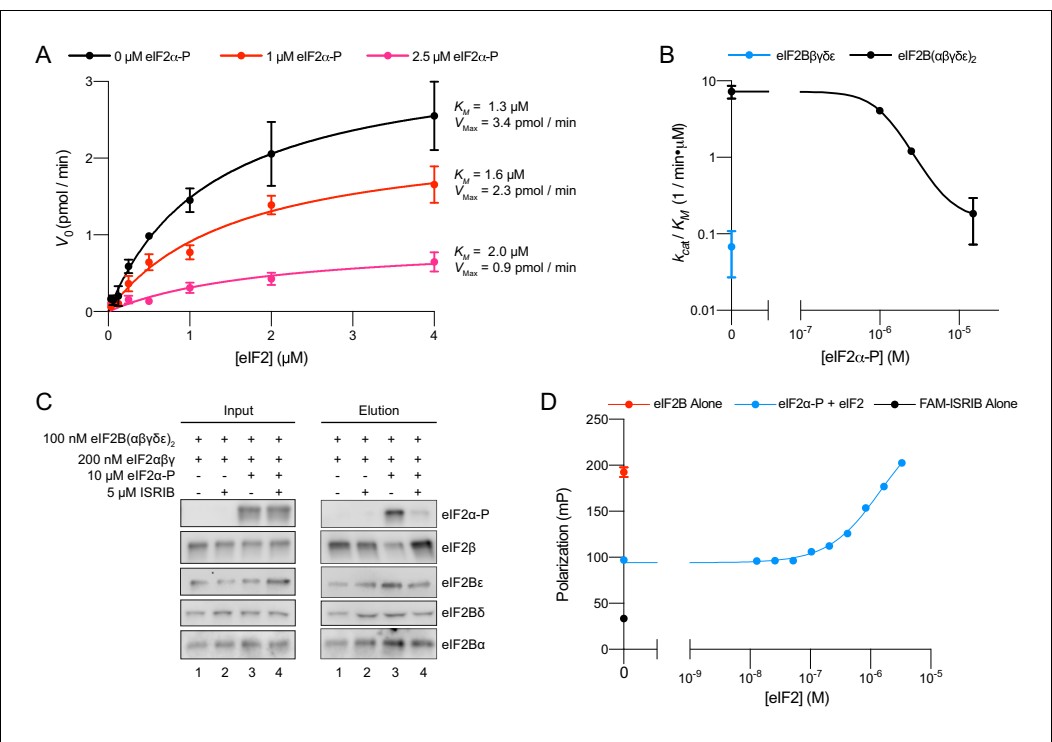

**Figure 6.** eIFα-P reduces eIF2B's catalytic activity and antagonizes eIF2 binding. (**A**) Initial velocity of eIF2B-catalyzed nucleotide exchange as a function of eIF2 concentration. eIF2B(αβδγε)$_2$ concentration was 10 nM. (**B**) Plot of $k_{cat}/K_M$ for tetramer and decamer at varying eIF2α-P concentrations, obtained by fitting the linear portion of the Michaelis–Menten saturation curve. Keeping the number of eIF2 binding sites constant, the eIF2B(αβδγε)$_2$ concentration was 10 nM while eIF2Bβδγε was 20 nM. (**C**) Western blot of purified protein recovered after incubation with eIF2B(αβδγε)$_2$ immobilized on anti-protein C antibody conjugated resin. eIF2Bα was protein C tagged. (**D**) Plot of fluorescence polarization signal before (black) and after incubation of FAM-ISRIB (2.5 nM) with 100 nM eIF2B(αβδγε)$_2$ (red) or 100 nM eIF2B(αβδγε)$_2$ + 6.0 μM eIF2α-P and varying concentrations of eIF2 (blue). For elution samples in (**C**), eIF2β, eIF2Bε, and eIF2Bα, and the eIF2Bδ and eIF2α-P blots are from the same gels, respectively. For input samples eIF2β and eIF2Bα, and the eIF2Bδ and eIF2α-P blots are from the same gels, respectively; eIF2Bε is from its own gel. Biological replicates: (**A, B**) n = 2; (**D**) n = 3. All error bars represent s.e.m. The online version of this article includes the following figure supplement(s) for figure 6:

**Figure supplement 1.** eIF2α-P decreases the initial velocity of eIF2's guanine nucleotide exchange factor (GEF) activity.

eIF2α-P inhibits the decamer by converting it to a tetramer-like state, rendering eIF2α-P-inhibited eIF2B decamers and eIF2B tetramers functionally equivalent (*Figure 6B, Figure 6—figure supplement 1*).

To further examine whether eIF2 and eIF2α-P antagonize one another's binding, we immobilized eIF2B decamers on agarose beads and incubated with combinations of eIF2, eIF2α-P, and ISRIB (*Figure 6C*). eIF2 readily bound to eIF2B with and without ISRIB (lanes 1 and 2) but eIF2α-P addition reduced the amount of eIF2 recovered (lane 3). As expected, ISRIB inhibited eIF2α-P binding and restored normal eIF2 binding (lane 4). Additionally, we utilized FAM-ISRIB as a tool to read out the eIF2-bound active state of eIF2B. Consistent with the data shown in *Figure 4E, F*, eIF2B addition to FAM-ISRIB increased polarization (*Figure 6D*, black and red data points, respectively), and FAM-ISRIB binding was blocked by the addition of eIF2α-P (blue data point on the y-axis). A titration of eIF2 into this reaction allowed FAM-ISRIB polarization to recover, indicating that eIF2 binds and disrupts eIF2α-P's inhibitory binding, which restores FAM-ISRIB binding. This result reinforces the notion that eIF2 and ISRIB binding are synergistic, that is, positively coupled.

## eIF2α-P inhibits eIF2B by inducing a conformational change

We next turned to structural studies to determine the basis of the decreased enzymatic activity and the apparent antagonism between eIF2α-P and both ISRIB and eIF2. First, we asked whether ISRIB binding alone causes a conformational change in decameric eIF2B. To this end, we prepared the apo-eIF2B decamer by combining eIF2Bβδγε tetramers and eIF2Bα$_2$ and subjected the sample to cryo-EM imaging. After 2D and 3D classification, we generated a single consensus structure of the apo-eIF2B decamer at 2.8 Å resolution (*Table 1*, *Figure 7—figure supplement 1*) with most side chains clearly resolved. This map allowed us to build an improved atomic model of the eIF2B decamer. This structure revealed that apo-eIF2B has an overall very similar structure as the ISRIB-bound decamer published previously (PDB ID: 6CAJ; *Tsai et al., 2018*; *Zyryanova et al., 2018*). Closer inspection revealed that ISRIB slightly draws the decamer's two halves together by comparison with the apostate but does not induce marked changes in eIF2B's overall conformation (*Figure 7—figure supplement 2A*).

We next examined the ISRIB binding pocket. In the apo versus the ISRIB-bound state, eIF2Bδ L179 shifts slightly into the pocket, occupying a position where it would clash with ISRIB binding, and eIF2Bβ H188 (a key ISRIB interactor) adopts a different rotamer (*Figure 7—figure supplement 2B*; *Tsai et al., 2018*). Overall, however, we conclude that ISRIB binding to the eIF2B decamer correlates with slight rearrangements that are primarily confined to the ISRIB binding pocket. Overlay of the apo decamer with structures of eIF2B bound to one or two copies of its enzymatically engaged substrate eIF2 also revealed unremarkable changes (*Kashiwagi et al., 2019*; *Kenner et al., 2019*; *Gordiyenko et al., 2019*; *Adomavicius et al., 2019*). We infer from these results that all of these structures represent, with the minor variations noted, the enzymatically active state of eIF2B, henceforth referred to as the 'A-State' ('A' for active).

By contrast, overlaying the eIF2B-eIF2α-P structure (PDB ID: 6O9Z) with the A-State structures revealed significant changes in the overall architecture of eIF2B (*Figure 7A*), henceforth referred to as the 'I-State' ('I' for inhibited) (*Kenner et al., 2019*). In the I-State, the two symmetrically opposed eIF2B tetramers have undergone a rocking motion that changes the angle between them by 7.5° (*Figure 7A*). The ISRIB pocket, consequentially, is lengthened by ~2 Å (*Figure 7B*). Critically, the substrate-binding cleft between eIF2Bβ and eIF2Bδ', where the N-terminal domain of the unphosphorylated eIF2α substrate binds, is widened by 2.6 Å, pulling IF4 away but leaving IF1–IF3 as available binding surfaces (*Figure 7C*, *Figure 7—figure supplement 3*). For both ISRIB and eIF2, these rearrangements break key anchoring interactions, providing a structural explanation why eIF2-P binding destabilizes ISRIB binding and compromises GEF activity. With only three of four interfaces available, eIF2 can still bind but would bind with lower affinity and may not necessarily be properly positioned, further explaining the reduced catalytic activity observed in *Figure 6A*. Conversely, in the A-State the cleft between eIF2Bα and eIF2Bδ' is widened by 5.5 Å (*Figure 7D*), disrupting the eIF2-P binding site and suggesting a possible mechanism for the antagonism between eIF2-P and eIF2/ISRIB.

Based on these structural comparisons, we conclude that eIF2B adopts at least two notably distinct conformational states, the A- and I-States. These two states are mutually exclusive (*Figure 8*). The A- and I-States, therefore, define an on-off switch of eIF2B's GEF activity and can be thought of

**Table 1.** Data collection, reconstruction, and model refinement statistics for the apo eIF2B decamer.

| Structure | Apo eIF2B decamer (PDB ID: 7L70; EMD-23209) |
|---|---|
| *Data collection* | |
| Microscope | Titan Krios |
| Voltage (keV) | 300 |
| Nominal magnification | ×105,000 |
| Exposure navigation | Image shift |
| Electron dose ($e^-Å^{-2}$) | 67 |
| Dose rate ($e^-$/pixel/s) | 8 |
| Detector | K3 summit |
| Pixel size (Å) | 0.835 |
| Defocus range (μm) | 0.6–2.0 |
| Micrographs | 1699 |
| *Reconstruction* | |
| Total extracted particles (no.) | 461,805 |
| Final particles (no.) | 198,362 |
| Symmetry imposed | C1 |
| FSC average resolution, masked (Å) | 3.8 |
| FSC average resolution, unmasked (Å) | 2.8 |
| Applied B-factor (Å) | 92.4 |
| Reconstruction package | Cryosparc 2.15 |
| *Refinement* | |
| Protein residues | 3156 |
| Ligands | 0 |
| RMSD bond lengths (Å) | 0.004 |
| RMSD bond angles (°) | 0.978 |
| Ramachandran outliers (%) | 0.06 |
| Ramachandran allowed (%) | 3.81 |
| Ramachandran favored (%) | 96.13 |
| Poor rotamers (%) | 2.61 |
| CaBLAM outliers (%) | 2.00 |
| Molprobity score | 1.83 |
| Clash score (all atoms) | 4.77 |
| B-factors (protein) | 88.43 |
| B-factors (ligands) | N/A |
| EMRinger score | 2.68 |
| Refinement package | Phenix 1.17.1-3660-000 |

FSC: Fourier shell correlation.

as functional equivalents to the decamer and tetramer assembly states, respectively. The A- to I-State transition thus appears to be the central mechanism underlying ISR activation.

## Discussion

As dysregulation of the ISR is increasingly implicated in numerous diseases with devastating consequences, understanding the mechanism of ISR signaling is of profound importance (*Costa-Mattioli and Walter, 2020*). The central ISR regulatory hub is the decameric guanine nucleotide

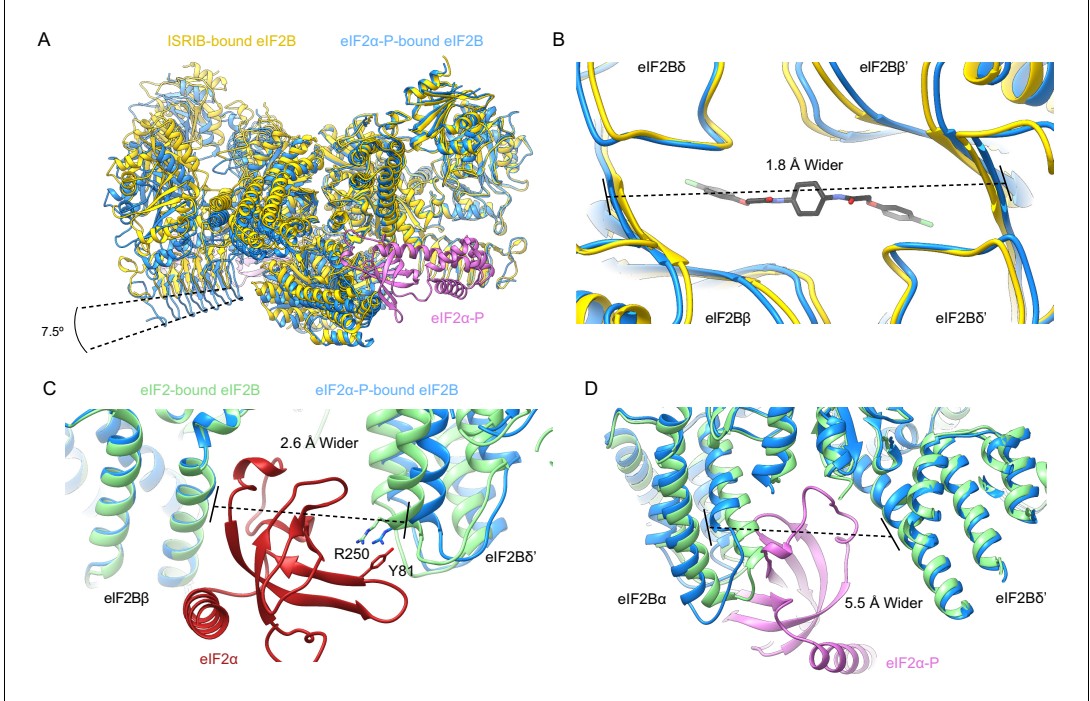

**Figure 7.** eIF2α-P binding conformationally inactivates eIF2B. (**A**) Overlay of the ISRIB-bound eIF2B structure (PDB ID: 6CAJ) to the eIF2α-P-bound eIF2B structure (PDB ID: 6O9Z). The 7.5° hinge movement between the two eIF2B halves was measured between the lines connecting eIF2Bε H352 and P439 in the ISRIB-bound versus eIF2α-P-bound structures. (**B**) Zoom-in view of the ISRIB binding pocket upon eIF2α-P binding. The ~2 Å pocket lengthening was measured between eIF2Bδ and eIF2Bδ' L482; the 'prime' to indicate the subunit of the opposing tetramer. ISRIB is shown in stick representation. (**C**) Overlay of eIF2-bound eIF2B (PDB ID: 6O85) and eIF2α-P-bound eIF2B. The 2.6 Å widening of the eIF2 binding site induced by eIF2α-P binding was measured between E139 and R250 of eIF2Bβ and eIF2Bδ', respectively. The side chains involved in the key cation–π interaction between R250 in eIF2Bδ and Y81 in eIF2α that is lost due to pocket expansion are shown. (**D**) Overlay of the eIF2-bound eIF2B to the eIF2α-P-bound eIF2B. The 5.5 Å narrowing of the eIF2α-P binding pocket causing a steric clash between eIF2Bα and eIF2α-P in the eIF2-bound state was measured between eIF2Bα S77 and eIF2Bδ L314. ISRIB-bound eIF2B is colored in gold, eIF2α-P-bound eIF2B in blue, and eIF2-bound eIF2B in light green. eIF2α-P is shown in pink and eIF2α in red. ISRIB is colored in CPK.

The online version of this article includes the following figure supplement(s) for figure 7:

**Figure supplement 1.** Cryo-electron microscopy workflow for apo-eIF2B decamer.

**Figure supplement 2.** ISRIB binding induces local pocket changes.

**Figure supplement 3.** eIF2-P binding pulls IF4 away but leaves IF1–IF3.

**Figure supplement 4.** Re-refinement of the ISRIB-bound eIF2B decamer.

exchange complex eIF2B, which activates eIF2 by loading it with GTP. Upon ISR activation in response to a variety of stress conditions, eIF2 becomes phosphorylated, converting it from eIF2B's substrate into an eIF2B inhibitor. Both eIF2 and eIF2-P are elongated protein complexes that contact eIF2B through multi-subunit, composite interaction surfaces (*Kenner et al., 2019*; *Kashiwagi et al., 2019*). The binding mode appears to be determined mainly by eIF2's α subunit, which anchors eIF2 and eIF2-P to their respective binding sites. For the substrate eIF2, binding aligns eIF2γ with eIF2B's catalytic site via IF1 and IF2 for nucleotide exchange. By contrast, for the inhibitor eIF2-P, binding positions its γ-subunit such that it could orthosterically prevent nonphosphorylated eIF2 substrate from engaging the catalytic machinery in eIF2Bε (*Kashiwagi et al., 2019*; *Kenner et al., 2019*). While this model was appealing based on the cryo-EM structures of eIF2B•eIF2-P complexes (*Kashiwagi et al., 2019*), the eIF2α C-terminal domain may retain sufficient flexibility to allow eIF2γ to avert the proposed clash (*Adomavicius et al., 2019*; *Ito et al., 2004*).

Expanding from this notion, in this work we show that allosteric rather than clash-based orthosteric competition contributes significantly to eIF2-P-mediated inhibition. We show that eIF2 and eIF2-P binding are negatively coupled, even when only the α subunit of eIF2-P is present. Thus, eIF2α-P binding impairs substrate binding even though the two binding sites are ~50 Å apart.

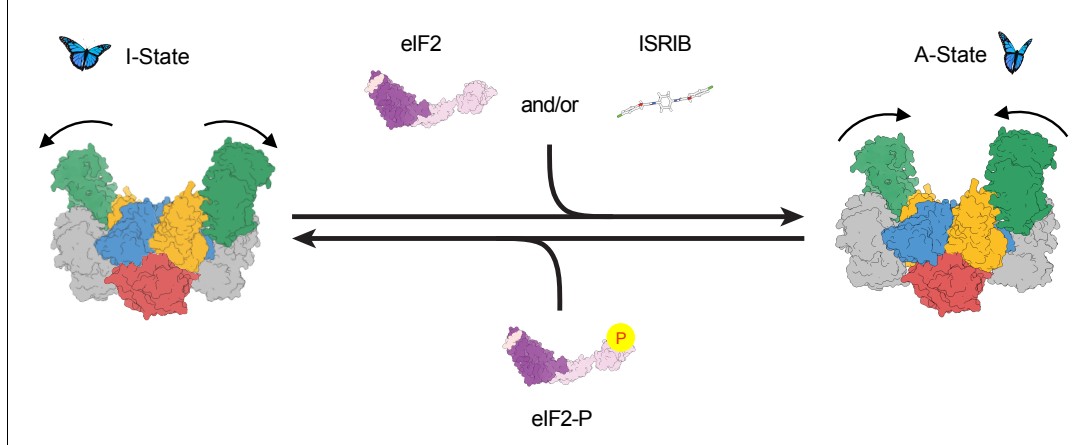

**Figure 8.** Model for modulation of eIF2B activity. ISRIB and eIF2 binding to eIF2B stabilize the active, 'wings up' conformation of eIF2B (A-State) while both eIF2-P (as well as eIF2α-P alone; not shown) stabilize the inactive 'wings down' conformation of eIF2B (I-State), which cannot engage ISRIB and exhibits reduced enzymatic activity and eIF2 binding (akin to an eIF2Bβδγε tetramer). As indicated by the structure of the apo eIF2B decamer, the conformational equilibrium in the absence of ligand likely favors the A-State, which is further stabilized by substrate eIF2 and/or ISRIB binding but antagonized by eIF2-P binding.

Further, the phosphorylated form of eIF2's α subunit alone inhibits GEF activity both through reduced substrate affinity and reduced eIF2B catalytic efficiency. Indeed, depending on the concentration regime, this change in eIF2B's intrinsic catalytic activity may be the main driver of lowered TC levels. With these data, we demonstrate that the eIF2γ subunit, which would be required for eIF2 inhibition via the clash-based orthosteric model, is mechanistically dispensable for eIF2-P's inhibitory role, although the added binding energy it contributes is certainly of importance in a cellular context.

Cryo-EM reconstructions support this model. They reveal a rocking motion of the two eIF2Bβδγε tetramers with eIF2Bα$_2$ acting as the fulcrum of the movement, akin to a butterfly raising and lowering its wings. These changes are induced by eIF2α-P alone. In the active or 'wings-up' A-State, eIF2Bβ and eIF2Bδ' subunits are sufficiently close to fully shape the eIF2α binding site, thus allowing nonphosphorylated substrate engagement. The A-State also contains a properly sized ISRIB binding pocket, thus rendering eIF2 and ISRIB binding synergistic. In contrast, the eIF2α-P binding site is misshapen and lacking properly positioned side chains critical for eIF2α-P binding. In the inhibited wings-down I-State, the eIF2α-P binding site is shaped correctly, while both the eIF2α (specifically IF4) and ISRIB binding sites are disrupted.

Prior to this work, models describing the molecular function of the drug-like small molecule ISRIB were exclusively focused on ISRIB's activity to promote eIF2B complex assembly. In vitro work from our and other labs demonstrated that eIF2Bβδγε tetramers assemble in the presence of ISRIB into eIF2B(βδγε)$_2$ octamers that approach the enzymatic activity of the eIF2B decamer, explaining how ISRIB could promote eIF2B assembly to restock the pool of active eIF2B when depleted by eIF2-P during ISR activation (*Tsai et al., 2018*; *Zyryanova et al., 2018*; *Sekine et al., 2015*; *Sidrauski et al., 2015*). However, because eIF2Bα$_2$ likewise has assembly-promoting activity, ISRIB can only exert this function when eIF2Bα$_2$ is limiting. We here validated this conjecture in living cells. Experimental depletion of eIF2Bα turned on ISR signaling in the absence of eIF2 phosphorylation, and ISRIB functionally substitutes for eIF2Bα$_2$. In the context of saturating eIF2Bα$_2$, we were thus left with a paradox regarding ISRIB's mechanism of action, which we resolve by showing that beyond a role in eIF2B assembly ISRIB antagonizes eIF2-P binding.

Previous works investigating the effects of compromising eIF2Bα (deletion, mutation, knockdown) did not report on eIF2B complex assembly and were predominantly performed in non-human model systems (*Pavitt et al., 1997*; *Hannig and Hinnebusch, 1988*; *Elsby et al., 2011*). Indeed, it is conceivable that eIF2B subcomplexes (and the role for these complexes in full heterodecamer assembly) are distinct between species. For example, in the fungus *Chaetomium thermophilum,* eIF2Bβ and eIF2Bδ appear to form heterotetrameric subcomplexes (*Kuhle et al., 2015*), whereas we see no

evidence for such stable assemblies in our work with human eIF2B. Thus, in other organisms enzymatically active octamers may form, and eIF2Bα's role may thus be primarily to allow eIF2-P binding. Another intriguing possibility is that in the long term, cells may enact mechanisms to compensate for the drop in TC levels that accompanies eIF2Bα depletion, consequent decamer disassembly, and decreased eIF2B GEF activity.

While our data clearly show that eIF2B is predominantly a decamer in K562 cells, this leaves open the possibility that the assembly state differs by cell type and/or is regulated physiologically. In principle, eIF2Bα could become limiting by regulation of its biosynthesis or degradation, post-translational modification, and/or sequestration into an unavailable pool. It is also important to note that an ISRIB-stabilized $eIF2B(\beta\delta\gamma\epsilon)_2$ octamer is inert to inhibition by eIF2-P. Such inhibition would require eIF2α-P to bind at the eIF2Bα/eIF2Bδ interface, which does not exist in complexes lacking eIF2Bα. We speculate that endogenous $eIF2B(\beta\delta\gamma\epsilon)_2$ octamers could be stabilized by putative alternate assembly factors, which could be metabolites or proteins that, like ISRIB, can substitute for $eIF2B\alpha_2$ in this regard.

In the course of this study, the demonstration that ISRIB still has a profound effect even in the context of fully assembled eIF2B led to the discovery of allosteric eIF2B regulation. While this manuscript was in preparation, a paper from Takuhiro Ito's and David Ron's laboratories was published that reached similar conclusions regarding ISRIB's effect on allosteric eIF2B regulation (*Zyryanova et al., 2021*). The work from these groups focuses almost exclusively on the allosteric effects promoted by the drug. Our results agree with their conclusions and demonstrate physiological significance. We show that substrate (eIF2) and inhibitor (eIF2-P) binding are negatively coupled. We additionally show that inhibitor binding reduces eIF2B's catalytic activity. Moreover, we show that by binding to the same binding site on eIF2B, ISRIB can affect the ISR in two modalities: (i) by promoting eIF2B assembly under conditions where $eIF2B\alpha_2$ is limiting or decamer stability may be compromised and (ii) biasing allosterically the conformational equilibrium of fully assembled decameric eIF2B towards the A-State, rendering inhibition by eIF2-P more difficult. Conceptually, these two modalities of ISRIB function are quite similar. In both cases, ISRIB promotes the completion of the eIF2α binding site by properly positioning IF4, so that it can cooperate with IF3 to anchor eIF2α. Indeed, in the I-State, the widening of the cleft between eIF2Bβ and eF2Bδ' effectively renders the available interaction surfaces on eIF2B equivalent to those on eIF2Bβδγε tetramers, limiting eIF2 engagement to IF1–IF3 as IF4 is pulled 'out of reach' as it would be in fully dissociated tetramers. In this way, we can think of eIF2B's I-State as 'conjoined tetramers' that remain tethered by $eIF2B\alpha_2$ but are functionally separate entities.

Considering the potential pharmacological applications of ISRIB, the relevant modality of ISRIB function may vary between different disease pathologies. In the case of vanishing white matter disease, for example, point mutations destabilize the eIF2B complex and ISRIB, therefore may provide primarily a stabilizing effect to recover eIF2B function (*Wong et al., 2018*). By contrast, in traumatic brain injury, sustained cognitive dysfunction is caused by persistent canonical ISR activation through eIF2-P (*Chou et al., 2017*). Hence, ISRIB would primarily counteract the aberrant ISR activation by predisposing eIF2B to the A-State. Other diseases are likely somewhere along the spectrum of purely assembly-based versus purely eIF2-P-based ISR activation. Our illustration of the differences between ISRIB's ability to resolve assembly-based stress versus eIF2-P-based stress should therefore inform how these different diseases are studied and ultimately treated.

The discovery of allosteric control of eIF2B activity raises intriguing possibilities. Indeed, we can envision that cell-endogenous modulators exist that work as activators (stabilizing the A-State) or inhibitors (stabilizing the I-State). Such putative ISR modulators could be small molecule metabolites or proteins and either bind to the ISRIB binding pocket or elsewhere on eIF2B to adjust the gain of ISR signaling to the physiological needs of the cell. Precedent for this notion comes from viruses that evolved proteins to counteract ISR-mediated antiviral defenses. The AcP10 protein in the Bw-CoV SW1 virus, for example, interacts with eIF2B to exert an ISRIB-like effect, likely predisposing eIF2B to the A-State (*Rabouw et al., 2020*). Regarding the observed changes in the ISRIB binding pocket, the newly gained structural insights can be applied to engineer novel pharmacological ISR modulators that may be effective in opening new therapeutic opportunities in different diseases.

## Materials and methods

### Cloning of tagged human eIF2B expression plasmids

*eIF2B2* (encoding eIF2Bβ) and *eIF2B4* (encoding eIF2Bδ) had previously been inserted into sites 1 and 2 of pACYCDuet-1, respectively (pJT073) (*Tsai et al., 2018*). In-Fusion HD cloning (Takarabio) was used to edit this plasmid further and insert mNeonGreen and a (GS)$_5$ linker at the C-terminus of *eIF2B2* and mScarlet-i and a (GS)$_5$ linker at the C-terminus of *eIF2B4* (pMS029). *eIF2B1* (encoding eIF2Bα) had previously been inserted into site 1 of pETDuet-1 (pJT075) (*Tsai et al., 2018*). In-Fusion HD cloning was used to edit this plasmid further and insert a protein C tag (EDQVDPRLIDGK) at the N-terminus of *eIF2B1*, immediately following the preexisting 6x-His tag (pMS027).

### Cloning of ATF4 and general translation reporter plasmids

The ATF4 translation reporter was generated using In-Fusion HD cloning. A gBlock containing the ATF4 UTR with both uORF1 and uORF2, ecDHFR, and mNeonGreen was inserted into the pHR vector backbone. The vector was additionally modified to contain a bGH poly(A) signal. The general translation reporter was similarly generated using a gBlock containing a modified ATF4 UTR with both uORF1 and uORF2 removed, ecDHFR, and mScarlet-i.

### Cloning of eIF2B homology-directed recombination (HDR) template plasmids

HDR template plasmids were generated using Gibson Assembly (NEB) cloning. gBlocks containing mNeonGreen and flanking *eIF2B2* homology arms (pMS074), mScarlet-i and flanking *eIF2B4* homology arms (pMS075), and FKBP12$^{F36V}$ and flanking *eIF2B1* homology arms (pMS101) were inserted into the pUC19 vector. Homology arms were 300 bp in all instances.

### ISR reporter cell line generation

K562 cells expressing dCas9-KRAB as previously generated were used as the parental line (*Gilbert et al., 2014*). In the ISR reporter cell line, the general translation reporter and the ATF4 reporter were integrated sequentially using a lentiviral vector. Vesicular stomatitis virus-G pseudo-typed lentivirus was prepared using standard protocols and 293METR packaging cells. Viral supernatants were filtered through a 0.45 μm (low protein binding) filter unit (EMD Millipore). The filtered retroviral supernatant was then concentrated twentyfold using an Amicon Ultra-15 concentrator (EMD Millipore) with a 100,000 Da molecular mass cutoff. Concentrated supernatant was then used the same day or frozen for future use. For spinfection, approximately 900,000 K562 cells were mixed with concentrated lentivirus + virus collection media (DMEM containing 4.5 g/l glucose supplemented with 10% FBS, 6 mM L-glutamine, 15 mM HEPES, and penicillin/streptomycin), supplemented with polybrene to 8 μg/ml, brought to 1.5 ml in a six-well plate, and centrifuged for 1.5 hr at 1000 *g*. Cells were then allowed to recover and expand for ~1 week before sorting on a Sony SH800 cytometer to isolate cells that had integrated the reporter. Before sorting, cells were treated with 20 μM trimethoprim for 3 hr to stabilize the general translation reporter product (ecDHFR-mScarlet-i). mScarlet-i-positive cells (targeting a narrow window around median reporter fluorescence) were then sorted into a final pooled population.

Integration of the ATF4 reporter was performed as above using the general translation reporter-containing cells as stock for spinfection. At the sorting stage, cells were again treated with 20 μM trimethoprim as well as 100 nM thapsigargin (tg) to allow ATF4 reporter translation to be monitored. The highest 3% of mNeonGreen-positive cells were sorted into a final pooled population.

The *eIF2B1* locus was endogenously edited using modifications to previous protocols (*Leonetti et al., 2016*). In brief, an HDR template was prepared by PCR amplifying from pMS101 using oligos oMS266 and oMS267 (*Table 2*). This product was then purified and concentrated to >1 μM using magnetic SPRI beads (Beckman Coulter). A 2.2 μl Cas9 buffer (580 mM KCl, 40 mM Tris pH 7.5, 2 mM TCEP (tris(20carboxyethyl)phosphine)-HCl, 2 mM MgCl$_2$, and 20% v/v glycerol) was added to 1.3 μl of 100 μM sgRNA (sgMS006, purchased from Synthego) and 2.9 μl H$_2$O and incubated at 70°C for 5 min. Then, 1.6 μl of 62.5 μM Alt-R S.p Cas9 Nuclease V3 (IDT) was slowly added to the mix and incubated at 37°C for 10 min. The donor template was then added to a final concentration of 0.5 μM, and final volume of 10 μl and the RNP mix was stored on ice.

**Table 2.** Oligos and sgRNAs.

| Oligo | Sequence | Use |
|---|---|---|
| oMS266 | /5InvddT/G*G*G*A*A*CCTCTTCTGTAACTCCTTAGC | Amplify HDR template |
| oMS267 | /5InvddT/C*C*T*G*A*G*GGCAAACAAGTGAGCAGG | Amplify HDR template |
| oMS269 | TCGTGCCAGCCCCCTAATCT | Validate eIF2Bα tagging |
| oMS270 | CTGAACGGCGCTGCTGTAGC | Validate eIF2Bα tagging |
| oMS256 | AGTGAACTCTACCATCCTGA | Validate eIF2Bβ tagging |
| oMS258 | TTAGGTGGACTCCTGTGC | Validate eIF2Bβ tagging |
| oMS096 | CTGGCTAACTGGCAGAACC | Validate eIF2Bδ tagging |
| oMS268 | AGAAACAAAGGCAGCAGAGT | Validate eIF2Bδ tagging |
| sgMS001 | CAATCTGCTTAGGACACGTG | Target Cas9 to eIF2BβC-terminus |
| sgMS004 | AGAGCAGTGACCAGTGACGG | Target Cas9 to eIF2Bδ C-terminus |
| sgMS006 | GAGGACGCCATGGACGACAA | Target Cas9 to eIF2βα N-terminus |

HDR: homology-directed recombination.

ISR reporter cells were treated with 200 ng/ml nocodazole (Sigma-Aldrich) to synchronize at G2/M phase for 18 hr. Approximately 200,000 cells were resuspended in a mixture of room temperature Amaxa solution (16.4 µl SF Solution, 3.6 µl Supplement [Lonza]). The cell/Amaxa solution mixture was added to the RNP mix and then pipetted into the bottom of a 96-well nucleofection plate (Lonza). This sample was then nucleofected using the 4D-Nucleofector Core unit and 96-well shuttle device (Lonza) with program FF-120. The cells were then returned to pre-warmed RPMI media in a 37°C incubator and allowed to recover/expand for >1 week. Limiting dilutions of cells were then prepared and plated in individual wells of a 96-well plate and allowed to grow up to identify clonal cells. Identification of edited clones was performed by western blotting for eIF2Bα and PCR amplification of the edited locus.

## FRET assembly state reporter cell line generation

eIF2Bβ-mNeonGreen-Flag-tagged cells were generated as described above with pMS074 used to PCR amplify the HDR template and sgMS001 used as the sgRNA. After recovery and expansion, the edited cells were sorted on a Sony SH800 cytometer, and the top 0.1% of mNeonGreen fluorescing cells were sorted into a polyclonal population. After expansion, recovery, and determining that the editing efficiency was over 90% in this population, the polyclonal cells were subjected to a second round of nucleofection using an HDR template amplified off of pMS075 to endogenously tag eIF2Bδ. sgMS004 was used to target the *eIF2B2* locus. Nucleofection conditions were as described above. After ~1 week of recovery and expansion, cells were again sorted as described above to isolate the highest mScarlet-i fluorescing cells. After ~1 week of recovery, limiting dilutions were prepared as described above to isolate and validate editing in individual clones. A fully *eIF2B2*-edited and *eIF2B4*-edited clone was then subjected to a third round of nucleofection to introduce the eIF2Bα-FKBP12$^{F36V}$ fusion. This was performed under identical conditions to those described above for the ISR reporter cell line.

## ATF4/general translation reporter assays

ISR reporter cells (at ~500,000/ml) were co-treated with varying combinations of drugs (trimethoprim, dTag13, thapsigargin, ISRIB) and incubated at 37°C until the appropriate timepoint had been reached. At this time, the plate was removed from the incubator and samples were incubated on ice for 10 min. Then ATF4 (mNeonGreen) and General Translation (mScarlet-i) reporter levels were read out using a high-throughput sampler attached to a BD FACSCelesta cytometer. Data was analyzed in FlowJo version 10.6.1, and median fluorescence values for both reporters were exported and plotted in GraphPad Prism 8. Where appropriate, curves were fit to log[inhibitor] versus response function with variable slope.

## In vivo FRET assembly state reporter assays

FRET assembly state reporter cells (at ~500,000/ml) were dosed with varying combinations of drugs (dTag13, thapsigargin, ISRIB) and incubated at 37°C until the appropriate timepoint had been reached. At this time, the plate was removed from the incubator, and samples were transferred to 5 ml FACS tubes. Samples were kept on ice. FRET signal was measured on a BD FACSAria Fusion cytometer. Data were analyzed in FlowJo version 10.6.1, and median fluorescence values for both mNeonGreen and mScarlet-i emission after mNeonGreen excitation were calculated. The ratio of these two values (termed 'FRET signal') was plotted in GraphPad Prism 8. Where appropriate, curves were fit to log[inhibitor] versus response function with variable slope.

## Western blotting

Approximately 1,000,000 cells of the appropriate cell type were drugged as described in individual assays and then pelleted (500× *g* for 4 min) at 4°C, resuspended in ice-cold phosphate-buffered saline (PBS), pelleted again, and then resuspended in 150 µl lysis buffer (50 mM Tris-HCl pH 7.4, 150 mM NaCl, 1 mM EDTA, 1% v/v Triton X-100, 10% v/v glycerol, 1× cOmplete protease inhibitor cocktail [Roche], and 1× PhosSTOP [Roche]). Cells were rotated for 30 min at 4°C and then spun at 12,000 g for 20 min to pellet cell debris. The supernatant was removed to a fresh tube, and protein concentration was measured using a bicinchoninic acid assay. Within an experiment, total protein concentration was normalized to the least concentrated sample (typically all values were within ~10% and in the 1 µg/µl range). A 5× Laemmli loading buffer (250 mM Tris-HCl pH 6.8, 30% glycerol, 0.25% bromophenol blue, 10% SDS, 5% β-mercaptoethanol) was added to each sample. Samples were placed in a 99°C heat block for 10 min. Equal protein content for each condition (targeting 10 µg) was run on 10% Mini-PROTEAN TGX precast protein gels (Bio-Rad). After electrophoresis, proteins were transferred onto a nitrocellulose membrane. Primary antibody/blocking conditions for each protein of interest are outlined in *Table 3*. Initial blocking is performed for 2 hr. Primary antibody staining was performed with gentle agitation at 4°C overnight. After washing four times in the appropriate blocking buffer, secondary antibody staining was performed for 1 hr at room temperature and then membranes were washed 3× with the appropriate blocking buffer, and then 1× with tris-buffered saline, 0.1% TWEEN (TBS-T) or phosphate-buffered saline, 0.1% TWEEN (PBS-T) as appropriate. Membranes were developed with SuperSignal West Dura (Thermo Fisher Scientific). Developed membranes were imaged on a LI-COR Odyssey gel imager for 0.5–10 min depending on band intensity.

## FLAG immunoprecipitation

Approximately 25,000,000 cells were drugged as described, removed from the incubator after 3 hr of treatment, and pelleted (3 min, 1000× *g*) then resuspended in ice-cold PBS, then pelleted again. Cells were then resuspended in 200 µl lysis buffer (25 mM HEPES pH 7.4, 150 mM KCl, 1% NP-40, 1 mM EDTA, 2.5× cOmplete protease inhibitor cocktail [Roche], and 1× PhosSTOP [Roche]). Cells were vortexed for 3 s then incubated on ice for 3 min, with this process repeated three times. Cell

**Table 3.** Antibodies for western blotting.

| Antibody target | Host | Dilution | Manufacturer | Blocking conditions |
|---|---|---|---|---|
| GAPDH | Rabbit | 1/2000 | Abcam | TBS-T + 3% BSA |
| eIF2Bα | Rabbit | 1/1000 | ProteinTech | TBS-T + 3% milk |
| eIF2Bβ | Rabbit | 1/1000 | ProteinTech | TBS-T + 3% milk |
| eIF2Bδ | Rabbit | 1/1000 | ProteinTech | TBS-T + 3% milk |
| eIF2Bε | Mouse | 1/1000 | Santa Cruz Biotechnology | PBS-T + 3% milk |
| ATF4 | Rabbit | 1/1000 | Cell Signaling | PBS-T + 3% milk |
| eIF2α-P | Rabbit | 1/1000 | Cell Signaling | PBS-T + 1% BSA |
| eIF2α | Rabbit | 1/1000 | Cell Signaling | PBS-T + 3% milk |
| eIF2β | Rabbit | 1/1000 | ProteinTech | PBS-T + 3% milk |

BSA: bovine serum albumin.

debris was pelleted as described above, and the supernatant was removed to a new tube. A portion was retained as the cell lysate fraction. The remaining cell lysate was incubated at 4°C overnight with M2 flag monoclonal antibody (Sigma-Aldrich) conjugated to magnetic Protein G Dynabeads (Invitrogen). Beads were washed 3× with 500 µl of sample buffer (20 mM HEPES pH 7.4, 100 mM KCl, 5 mM MgCl$_2$, and 1 mM TCEP) and then eluted using FLAG peptide at 200 µg/ml (eIF2B-bound fraction). Both fractions were then treated as described above for western blotting.

## gDNA isolation, PCR, and DNA gel of edited loci

gDNA from parental and edited cells was isolated using the PureLink Genomic DNA Mini Kit (Invitrogen) as per manufacturer's instructions. The targeted *EIF2B1*, *EIF2B2*, and *EIF2B4* loci were amplified with the primer pairs detailed in *Table 2* and run on a 1% agarose gel and imaged using a ChemiDoc XRS+ imaging system (Bio-Rad). The expected WT fragment lengths for the *EIF2B1*, *EIF2B2*, and *EIF2B4* products are 256, 151, and 224 bp, respectively, while the edited products are expected at 643, 955, and 997 bp, respectively.

## Purification of human eIF2B subcomplexes

Human eIFBα$_2$ (pJT075 or pMS027), eIF2Bβγδε (pJT073 and pJT074 co-expression), and eIF2Bβγδε-F (pMS029 and pJT074 co-expression) were purified as previously described (*Tsai et al., 2018*). All eIF2B(αβγδε)$_2$ used throughout was assembled by mixing purified eIF2Bβγδε and eIF2Bα$_2$ at the appropriate molar ratios.

## Purification of human eIF2α and eIF2α-P

The purification of human eIF2α was modified from a previous protocol (*Kenner et al., 2019*). Briefly, the expression plasmid for N-terminally 6x-His-tagged human eIF2α, pAA007, was heat-transformed into One Shot BL21 Star (DE3) chemically competent *E. coli* cells (Invitrogen), along with the tetracycline-inducible, chloramphenicol-resistant plasmid, pG-Tf2, containing the chaperones groES, groEL, and Tig (Takara Bio). Transformed cells were selected for in LB with kanamycin and chloramphenicol. When the culture reached an OD$_{600}$ of ~0.2, 1 ng/ml, tetracycline was added to induce expression of chaperones. At an OD$_{600}$ of ~0.8, the culture was cooled to room temperature, eIF2α expression was induced with 1 mM IPTG (Gold Biotechnology), and the culture was grown for 16 hr at 16°C. Cells were harvested and lysed through three cycles of high-pressure homogenization using the EmulsiFlex-C3 (Avestin) in a buffer containing 100 mM HEPES-KOH, pH 7.5, 300 mM KCl, 2 mM dithiothreitol (DTT), 5 mM MgCl$_2$, 5 mM imidazole, 10% glycerol, 0.1% IGEPAL CA-630, and cOmplete EDTA-free protease inhibitor cocktail (Roche). The lysate was clarified at 30,000× *g* for 30 min at 4°C. Subsequent purification steps were conducted on the ÄKTA Pure (GE Healthcare) system at 4°C. Clarified lysate was loaded onto a 5 ml HisTrap FF Crude column (GE Healthcare), washed in a buffer containing 20 mM HEPES-KOH, pH 7.5, 100 mM KCl, 5% glycerol, 1 mM DTT, 5 mM MgCl$_2$, 0.1% IGEPAL CA-630, and 20 mM imidazole, and eluted with 75 ml linear gradient of 20–500 mM imidazole. The eIF2α-containing fractions were collected and applied to a MonoQ HR 10/100 GL column (GE Healthcare) equilibrated in a buffer containing 20 mM HEPES-KOH pH 7.5, 100 mM KCl, 1 mM DTT, 5% glycerol, and 5 mM MgCl$_2$ for anion exchange. The column was washed in the same buffer, and the protein was eluted with an 80 ml linear gradient of 100 mM to 1 M KCl. eIF2α-containing fractions were collected and concentrated with an Amicon Ultra-15 concentrator (EMD Millipore) with a 30,000 Da molecular mass cutoff, spun down for 10 min at 10,000 g to remove aggregates. The supernatant was then chromatographed on a Superdex 75 10/300 GL (GE Healthcare) column equilibrated in a buffer containing 20 mM HEPES-KOH pH 7.5, 100 mM KCl, 1 mM DTT, 5 mM MgCl$_2$, and 5% glycerol, and concentrated using Amicon Ultra-15 concentrators (EMD Millipore) with a 10,000 Da molecular mass cutoff.

For the purification of human phosphorylated eIF2α (eIF2α-P), the protein was expressed and purified as described above for eIF2α, except that before size exclusion on the Superdex 75, the pooled anion exchange fractions were phosphorylated in vitro overnight at 4°C with 1 mM ATP and 1 µg of PKR$_{(252-551)}$-GST enzyme (Thermo Scientific) per milligram of eIF2α. Complete phosphorylation was confirmed by running the samples on a 12.5% Super-Sep PhosTag gel (Wako Chemicals).

## Purification of heterotrimeric human eIF2 and eIF2-P

Human eIF2 was prepared from an established recombinant *Saccharomyces cerevisiae* expression protocol (*de Almeida et al., 2013*). In brief, the yeast strain GP6452 (gift from the Pavitt Lab, University of Manchester) containing yeast expression plasmids for human eIF2 subunits and a deletion of GCN2 encoding the only eIF2 kinase in yeast was grown to saturation in synthetic complete media (Sunrise Science Products) with auxotrophic markers (-Trp, -Leu, -Ura) in 2% dextrose. The β and α subunits of eIF2 were tagged with 6x-His and FLAG epitopes, respectively. A 12 l yeast culture was grown in rich expression media containing yeast extract, peptone, 2% galactose, and 0.2% dextrose. Cells were harvested and resuspended in lysis buffer (100 mM Tris, pH 8.5, 300 mM KCl, 5 mM MgCl$_2$, 0.1% NP-40, 5 mM imidazole, 10% glycerol [Thermo Fisher Scientific], 1 mM TCEP, 1× cOmplete protease inhibitor cocktail [Sigma-Aldrich], 1 µg/ml each aprotinin [Sigma-Aldrich], leupeptin [Sigma-Aldrich], pepstatin A [Sigma-Aldrich]). Cells were lysed in liquid nitrogen using a steel blender. The lysate was centrifuged at 30,000× *g* for 30 min at 4°C. Subsequent purification steps were conducted on the ÄKTA Pure (GE Healthcare) system at 4°C. Lysate was applied to a 5 ml HisTrap FF Crude column (GE Healthcare) equilibrated in buffer (100 mM HEPES-KOH, pH 7.5, 100 mM KCl, 5 mM MgCl$_2$, 0.1% NP-40, 5% glycerol, 1 mM TCEP, 0.5× cOmplete protease inhibitor cocktail, 1 µg/ml each aprotinin, leupeptin, pepstatin A). eIF2 bound to the column was washed with equilibration buffer and eluted using a 50 ml linear gradient of 5–500 mM imidazole. Eluted eIF2 was incubated with FLAG M2 magnetic affinity beads, washed with FLAG wash buffer (100 mM HEPES-KOH, pH 7.5, 100 mM KCl, 5 mM MgCl$_2$, 0.1% NP-40, 5% glycerol, 1 mM TCEP, 1× cOmplete protease inhibitor cocktail, 1 µg/ml each aprotinin, leupeptin, pepstatin A) and eluted with FLAG elution buffer (identical to FLAG wash buffer but also containing 3× FLAG peptide [100 µg/ml, Sigma-Aldrich]). Protein was flash-frozen in liquid nitrogen and stored in elution buffer at −80°C.

For the purification of eIF2-P, the protein was purified as above, except that a final concentration of 10 nM recombinant PKR (Life Technologies # PV4821) and 1 mM ATP was added during incubation with FLAG M2 magnetic beads. These components were removed during the wash steps described above. Phosphorylation of the final product was verified by 12.5% SuperSep PhosTag gel (Wako Chemical Corporation).

Additional human eIF2 was purified as previously described with the only modification in one purification being an additional Avi-Tag on the eIF2α subunit (*Wong et al., 2018*). This material was a generous gift of Carmela Sidrauski and Calico Life Sciences.

## In vitro eIF2/eIF2α-P immunoprecipitation

eIF2B(αβδγε)$_2$ decamers were assembled by mixing eIF2Bβγδε and protein C-tagged eIF2Bα$_2$ in a 2:1 molar ratio and incubating at room temperature for at least 1 hr. Varying combinations of purified eIF2, eIF2α-P, eIF2B(αβδγε)$_2$, and ISRIB were incubated (with gentle rocking) with anti-protein C antibody conjugated resin (generous gift from Aashish Manglik) in assay buffer (20 mM HEPES-KOH, pH 7.5, 150 mM KCl, 5 mM MgCl$_2$, 1 mM TCEP, 1 mg/ml bovine serum albumin [BSA], 5 mM CaCl$_2$). After 1.5 hr, the resin was pelleted by benchtop centrifugation and the supernatant was removed. The resin was washed 3× with 1 ml of ice-cold assay buffer before it was resuspended in elution buffer (assay buffer with 5 mM EDTA and 0.5 mg/ml protein C peptide added) and incubated with gentle rocking for 1 hr. The resin was then pelleted and the supernatant was removed. Samples were analyzed by western blotting as previously described.

## Analytical ultracentrifugation

Analytical ultracentrifugation sedimentation velocity experiments were performed as previously described (*Tsai et al., 2018*).

## In vitro FRET assays

Equilibrium measurements of eIF2B assembly state were performed in 20 µl reactions with 50 nM eIF2Bβγδε-F + ISRIB or eIF2Bα$_2$ titrations in FP buffer (20 mM HEPES-KOH pH 7.5, 100 mM KCl, 5 mM MgCl$_2$, 1 mM TCEP) and measured in 384 square-well black-walled, clear-bottom polystyrene assay plates (Corning). Measurements were taken using the ClarioStar PLUS plate reader (BMG Lab-Tech) at room temperature. mNeonGreen was excited (470 nm, 8 nm bandwidth), and mNeonGreen (516 nm, 8 nm bandwidth) and mScarlet-i (592 nm, 8 nm bandwidth) emission were monitored. FRET

signal ($E_{592}/E_{516}$) is the ratio of mScarlet-i emission after mNeonGreen excitation and mNeonGreen emission after mNeonGreen excitation. All reactions were performed in a final 0.5% DMSO content. Samples were incubated for 1 hr before measurement. Data were plotted in GraphPad Prism 8, and curves were fit to log(inhibitor) versus response function with variable slope.

Kinetic measurements of eIF2B assembly were performed in the same final volume and buffer as above. Then, 10 µl of 2× ISRIB, eIF2Bα$_2$, or ISRIB + eIF2Bα$_2$ stocks were placed in wells of the above-described assay plate. Also, 10 µl of 100 nM (2×) eIF2Bβγδε-F was then added and mixed with the contents of each well using a 20 µl 12-channel multichannel pipette. Measurements were taken using the above instrument every 18 s for the first 24 cycles and then every 45 s for the next 60 cycles. mNeonGreen was excited (470 nm, 16 nm bandwidth), and mNeonGreen (516 nm, 16 nm bandwidth) and mScarlet-i (592 nm, 16 nm bandwidth) emission were monitored. After this association phase, 18 µl was removed from each well using a multichannel pipette and mixed with 1 µl of 20 µM (20×) untagged eIF2Bβγδε pre-loaded into PCR strips. The material was then returned to the original wells and measurement of dissociation began. Measurements were taken every 18 s for the first 24 cycles and then every 45 s for the next 120 cycles. Data were plotted in GraphPad Prism 8. Association and dissociation phases were fit separately using the one-phase association and dissociation–one phase exponential decay models, respectively. Global fits were performed on the ISRIB titrations or eIF2Bα$_2$ titrations. When modeling dissociation, the median buffer signal at assay completion was used to set the bottom baseline for conditions where full dissociation was not observed (eIF2Bα$_2$ and eIF2Bα$_2$ + ISRIB conditions).

## GDP exchange assay

In vitro detection of GDP binding to eIF2 was adapted from a published protocol for a fluorescence intensity–based assay describing dissociation of eIF2 and nucleotide (*Sekine et al., 2015*). We first performed a loading assay for fluorescent BODIPY-FL-GDP as described (*Tsai et al., 2018*). Purified eIF2 (100 nM) was incubated with 100 nM BODIPY-FL-GDP (Thermo Fisher Scientific) in assay buffer (20 mM HEPES-KOH, pH 7.5, 100 mM KCl, 5 mM MgCl$_2$, 1 mM TCEP, and 1 mg/ml BSA) to a volume of 18 µl in 384 square-well black-walled, clear-bottom polystyrene assay plates (Corning). The GEF mix was prepared by incubating a 10× solution of eIF2B(αβγδε)$_2$ with 10× solutions of eIF2-P or eIF2α-P. For analyzing the effect of ISRIB, the 10× GEF mixes were pre-incubated with 2% NMP or 10 µM ISRIB in N-methyl-2-pyrrolidone (NMP), such that the final NMP and ISRIB concentration was 1 µM and the final NMP concentration was 0.2%. To compare nucleotide exchange rates, the 10× GEF mixes were spiked into the 384-well plate wells with a multichannel pipette, such that the resulting final concentration of eIF2B(αβγδε)$_2$ was 10 nM, and the final concentration of other proteins and drugs are as indicated in the figures. Subsequently, in the same wells, we performed a 'GDP unloading assay' as indicated in the figures. After completion of the loading reaction, wells were next spiked with 1 mM GDP to start the unloading reaction at t = 0. Fluorescence intensity was recorded every 10 s for 60 min using a Clariostar PLUS (BMG LabTech) plate reader (excitation wavelength: 497 nm, bandwidth 14 nm; emission wavelength: 525 nm, bandwidth: 30 nm). Data collected were fit to a first-order exponential.

## Michaelis–Menten kinetics

BODIPY-FL-GDP loading assays were performed as described above, varying substrate concentration in twofold increments from 31.25 nM to 4 µM while eIF2B decamer concentration was held constant at 10 nM. Experiments containing tetramer were performed at 20 nM, such that the number of active sites was held constant. For conditions reported in *Figure 6A*, initial velocity was determined by a linear fit to timepoints acquired at 5 s intervals from 50 to 200 s after addition of GEF. For eIF2B tetramer and eIF2B decamer + 15 µM eIF2α-P conditions, timepoints were acquired at 20 s intervals and initial velocity was determined by a linear fit to timepoints 400–1000 s. $k_{cat}$ and $K_M$ were determined by fitting the saturation curves shown in *Figure 6A* to the Michaelis–Menten equation. Data collected for tetramer and decamer + 15 µM eIF2α-P conditions fell within the linear portion of the Michaelis–Menten saturation curve, and thus the linear portion of each curve was fit to determine the $k_{cat}$ / $K_M$ values reported in *Figure 6B*.

## FAM-ISRIB binding assay

All fluorescence polarization measurements were performed in 20 µl reactions with 100 nM eIF2B ($\alpha\beta\gamma\delta\epsilon$)$_2$ + 2.5 nM FAM-ISRIB (Praxis Bioresearch) in FP buffer (20 mM HEPES-KOH pH 7.5, 100 mM KCl, 5 mM MgCl$_2$, 1 mM TCEP) and measured in 384-well non-stick black plates (Corning 3820) using the ClarioStar PLUS (BMG LabTech) at room temperature. Prior to reaction setup, eIF2B ($\alpha\beta\gamma\delta\epsilon$)$_2$ was assembled in FP buffer using eIF2B$\beta\gamma\delta\epsilon$ and eIF2B$\alpha_2$ in 2:1 molar ratio for at least 15 min at room temperature. FAM-ISRIB was always first diluted to 2.5 µM in 100% NMP prior to dilution to 50 nM in 2% NMP and then added to the reaction. For titrations with eIF2, eIF2-P, eIF2$\alpha$, and eIF2$\alpha$-P, dilutions were again made in FP buffer, and the reactions with eIF2B, FAM-ISRIB, and these dilutions were incubated at 22°C for 30 min prior to measurement of parallel and perpendicular intensities (excitation: 482 nm; emission: 530 nm). To measure the effect of phosphorylated eIF2 on FAM-ISRIB binding to eIF2B, we additionally added 1 µl (0.21 µg) of PKR$_{(252-551)}$-GST enzyme (Thermo Scientific) and 1 mM ATP to the reaction with eIF2B, FAM-ISRIB, and eIF2 before incubation at 22°C for 30 min. For the measurement of eIF2 and eIF2$\alpha$-P competition, 19 µl reactions of 100 nM eIF2B($\alpha\beta\gamma\delta\epsilon$)$_2$, 2.5 nM FAM-ISRIB, and 6 µM eIF2$\alpha$-P were incubated with titrations of eIF2 for 30 min before polarization was measured. To confirm that FAM-ISRIB binding was specific to eIF2B, after each measurement, ISRIB was spiked to 1 µM into each reaction (from a 40 µM stock in 100% NMP), reactions were incubated for 15 min at 22°C, and polarization was measured again using the same gain settings. Data were plotted in GraphPad Prism 8, and where appropriate, curves were fit to log[inhibitor] versus response function with variable slope.

The kinetic characterization of FAM-ISRIB binding during eIF2$\alpha$ phosphorylation was assayed in 19 µl reactions of 100 nM eIF2B($\alpha\beta\gamma\delta\epsilon$)$_2$, 2.5 nM FAM-ISRIB, 1 mM ATP, and 5.6 µM eIF2$\alpha$/eIF2$\alpha$-P in FP buffer. These solutions were pre-incubated at 22°C for 30 min before polarization was measured every 15 s (30 flashes/s). After four cycles, 1 µl (0.21 µg) of PKR$_{(252-551)}$-GST enzyme (Thermo Scientific) was added, and measurement was resumed. Dephosphorylation reactions were set up in an analogous way, but instead of ATP 1 mM MnCl$_2$ was added and 1 µl (400 U) of $\lambda$ phosphatase (NEB) was used instead of PKR.

## Sample preparation for cryo-EM

Decameric eIF2B($\alpha\beta\gamma\delta\epsilon$)$_2$ was prepared by incubating 20 µM eIF2B$\beta\gamma\delta\epsilon$ with 11 µM eIF2B$\alpha_2$ in a final solution containing 20 mM HEPES-KOH, 200 mM KCl, 5 mM MgCl$_2$, and 1 mM TCEP. This 10 µM eIF2B($\alpha\beta\gamma\delta\epsilon$)$_2$ sample was further diluted to 750 nM and incubated on ice for 1 hr before plunge freezing. A 3 µl aliquot of the sample was applied onto the Quantifoil R 1.2/1/3 400 mesh Gold grid and waited for 30 s. A 0.5 µl aliquot of 0.1–0.2% Nonidet P-40 substitute was added immediately before blotting. The entire blotting procedure was performed using Vitrobot (FEI) at 10°C and 100% humidity.

## Electron microscopy data collection

Cryo-EM data for the *apo* decamer of eIF2B was collected on a Titan Krios transmission electron microscope operating at 300 keV, and micrographs were acquired using a Gatan K3 direct electron detector. The total dose was 67 e$^-$/ Å$^2$, and 117 frames were recorded during a 5.9 s exposure. Data was collected at $\times$105,000 nominal magnification (0.835 Å/pixel at the specimen level), and nominal defocus range of −0.6 to −2.0 µm. Cryo-EM data for the ISRIB-bound eIF2B decamer (EMDB:7442, 7443, and 7444) (*Tsai et al., 2018*) and the eIF2-bound eIF2B decamer were collected as described previously (EMDB:0651) (*Kenner et al., 2019*).

## Image processing

For the apo decamer, the micrograph frames were aligned using MotionCorr2 (*Zheng et al., 2017*). The contrast transfer function (CTF) parameters were estimated with GCTF (*Zhang, 2016*). Particles were automatically picked using Gautomatch and extracted in RELION using a 400-pixel box size (*Scheres, 2012*). Particles were classified in 2D in Cryosparc (*Punjani et al., 2017*). Classes that showed clear protein features were selected and extracted for heterogeneous refinement using the ISRIB-bound decamer as a starting model (EMDB ID: 7442) (*Tsai et al., 2018*). Homogeneous refinement was performed on the best model to yield a reconstruction of 2.89 Å. Nonuniform refinement was then performed to yield a final reconstruction of 2.83 Å. For the ISRIB-bound eIF2B decamer

(EMDB:7442, 7443, and 7444) (*Tsai et al., 2018*) and the eIF2-bound eIF2B decamer (EMDB:0651) (*Kenner et al., 2019*), the published maps were used for further model refinement.

### Atomic model building, refinement, and visualization

For all models, previously determined structures of the human eIF2B complex (PDB: 6CAJ [*Tsai et al., 2018*]), human eIF2α (PDBs: 1Q8K [*Ito et al., 2004*] and 1KL9 [*Nonato et al., 2002*]), the C-terminal HEAT domain of eIF2Bε (PDB: 3JUI [*Wei et al., 2010*]), and mammalian eIF2γ (PDB: 5K0Y [*Esser et al., 2017*]) were used for initial atomic interpretation. The models were manually adjusted in Coot (*Emsley and Cowtan, 2004*) or ISOLDE (*Croll, 2018*) and then refined in phenix.

**Table 4.** Data collection, reconstruction and refinement statistics for the ISRIB-bound eIF2B decamer.

| Structure | ISRIB-bound eIF2B decamer from Janelia (PDB ID: 6CAJ) (*Tsai et al., 2018*) | ISRIB-bound eIF2B decamer from Berkeley (PDB ID: 6CAJ) (*Tsai et al., 2018*) |
|---|---|---|
| *Data collection* | | |
| Voltage (keV) | 300 | 300 |
| Nominal magnification | ×29,000 | ×29,000 |
| Per frame electron dose (e$^-$Å$^{-2}$) | 1.19 | 1.63 |
| Spherical aberration (mm) | 2.7 | 2.62 |
| Number of frames | 67 | 27 |
| Detector | K2 summit | K2 summit |
| Pixel size (Å) | 1.02 | 0.838 |
| Defocus range (μm) | −0.3 to −3.9 | −0.3 to −3.9 |
| Micrographs | 1780 | 1515 |
| Frame length (s) | 0.15 | 0.18 |
| Detector pixel size (μm) | 5.0 | 5.0 |
| *Reconstruction using particles from both data sets after magnification rescaling* | | |
| Particles following 2D classification | 202,125 | |
| FSC average resolution unmasked (Å) | 3.4 | |
| FSC average resolution masked (Å) | 3.0 | |
| Map sharpening B-factor | −60 | |
| *Refinement* PDB ID: 7L7G (update to 6CAJ); EMD-7443 | | |
| Protein residues | 3198 | |
| Ligands | 1 | |
| RMSD bond lengths (Å) | 0.004 | |
| RMSD bond angles (°) | 0.967 | |
| Ramachandran outliers (%) | 0.00 | |
| Ramachandran allowed (%) | 5.40 | |
| Ramachandran favored (%) | 94.60 | |
| Poor rotamers (%) | 1.00 | |
| Molprobity score | 1.81 | |
| Clash score (all atoms) | 7.95 | |
| B-factors (protein) | 65.93 | |
| B-factors (ligands) | 52.57 | |
| EMRinger score | 2.37 | |
| Refinement package | Phenix 1.17.1-3660-000 | |

FSC: Fourier shell correlation.

real_space_refine (*Adams et al., 2010*) using global minimization, secondary structure restraints, Ramachandran restraints, and local grid search. Then iterative cycles of manually rebuilding in Coot and phenix.real_space_refine with additional B-factor refinement were performed. The final model statistics were tabulated using Molprobity (*Tables 1* and *4*; *Chen et al., 2010*). Map versus atomic model FSC plots were computed after masking using Phenix validation tools. Distances and rotations were calculated from the atomic models using UCSF Chimera. Final atomic models have been deposited at the PDB with the following accession codes: ISRIB-bound eIF2B (6caj, updated), eIF2•eIF2B•ISRIB (6o85); and apo eIF2B (7L70). Molecular graphics and analyses were performed with the UCSF Chimera package (*Pettersen et al., 2004*). UCSF Chimera is developed by the Resource for Biocomputing, Visualization, and Informatics and supported by NIGMS P41-GM103311.

## Acknowledgements

We thank the Walter Lab for helpful discussions throughout the course of this project; G Narlikar for insight into kinetic analyses; the labs of A Manglik, M Kampmann, and J Weissman for shared reagents; C Sidrauski and Calico for a generous gift of purified eIF2 heterotrimer; Z Yu and D Bulkley of the UCSF Center for Advanced Cryo-EM facility, which is supported by NIH grants S10OD021741 and S10OD020054 and the Howard Hughes Medical Institute (HHMI). We also thank the QB3 shared cluster for computational support. This work was supported by generous support from Calico Life Sciences LLC (to PW); a generous gift from The George and Judy Marcus Family Foundation (to PW); the Belgian-American Educational Foundation (BAEF) Postdoctoral Fellowship (to MB), the Damon Runyon Cancer Research Foundation Postdoctoral fellowship (to LW); the Jane Coffin Child Foundation Postdoctoral Fellowship (to RL); a Chan Zuckerberg Biohub Investigator award; and an HHMI Faculty Scholar grant (AF). PW is an investigator of the Howard Hughes Medical Institute.

## Additional information

### Competing interests

Adam Frost: Reviewing editor, *eLife*. Peter Walter: PW is an inventor on U.S. Patent 9708247 held by the Regents of the University of California that describes ISRIB and its analogs. Rights to the invention have been licensed by UCSF to Calico. The other authors declare that no competing interests exist.

### Funding

| Funder | Grant reference number | Author |
| --- | --- | --- |
| Howard Hughes Medical Institute | Investigator Grant | Peter Walter |
| Howard Hughes Medical Institute | HHMI Faculty Scholar Grant | Adam Frost |
| Calico Life Sciences LLC | | Peter Walter |
| The George and Judy Marcus Family Foundation | | Peter Walter |
| Damon Runyon Cancer Research Foundation | Postdoctoral Fellowship | Lan Wang |
| Jane Coffin Childs Memorial Fund for Medical Research | Postdoctoral Fellowship | Rosalie Lawrence |
| Belgian American Educational Foundation | Postdoctoral Fellowship | Morgane Boone |
| Chan Zuckerberg Initiative | Investigator Grant | Adam Frost |

The funders had no role in study design, data collection and interpretation, or the decision to submit the work for publication.

## Author contributions
Michael Schoof, Conceptualization, Formal analysis, Investigation, Writing - original draft, Writing - review and editing; Morgane Boone, Rosalie Lawrence, Formal analysis, Investigation, Writing - review and editing; Lan Wang, Formal analysis, Investigation, Visualization, Writing - review and editing; Adam Frost, Formal analysis, Supervision, Funding acquisition, Investigation, Visualization, Writing - review and editing; Peter Walter, Conceptualization, Supervision, Funding acquisition, Writing - original draft, Project administration, Writing - review and editing

## Author ORCIDs
Michael Schoof (iD) https://orcid.org/0000-0003-3531-5232
Morgane Boone (iD) https://orcid.org/0000-0002-7807-5542
Lan Wang (iD) http://orcid.org/0000-0002-8931-7201
Adam Frost (iD) https://orcid.org/0000-0003-2231-2577
Peter Walter (iD) https://orcid.org/0000-0002-6849-708X

## Decision letter and Author response
Decision letter https://doi.org/10.7554/eLife.65703.sa1
Author response https://doi.org/10.7554/eLife.65703.sa2

## Additional files
### Supplementary files
• Transparent reporting form

### Data availability
All data generated or analysed during this study are included in the manuscript and supporting files.

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
