## [Decision Letter]

**Acceptance summary:**

The authors have combined several approaches to study the mechanism of action of the drug-like compound, ISRIB (Integrated Stress Response Inhibitor). The major conceptual advance of this work is the discovery that ISRIB can inhibit the integrated stress response using two different modalities. This study and the recently published work from Zyryanova et al. in Molecular Cell (2021) complement one another and provide clear insights into how ISRIB confers resistance to the phosphorylation of initiation factor 2 (eIF2) and downstream effects on the Integrated Stress Response.

**Decision letter after peer review:**

Thank you for submitting your article "eIF2B Conformation and Assembly State Regulates the Integrated Stress Response" for consideration by *eLife*. Your article has been reviewed by three peer reviewers, one of whom is a member of our Board of Reviewing Editors, and the evaluation has been overseen by Suzanne Pfeffer as the Senior Editor. The reviewers have opted to remain anonymous.

Essential Revisions:

The reviewers considered your work of interest and worthy of publication in *eLife* after revisions in response to their criticisms. The reviewers have provided many thoughtful and constructive comments. We are therefore enclosing the full reviews, but we are listing below the comments which ought to be addressed, as follows:

1) It is important to calculate a K1/2 constant of association/inhibition concentrations for the plots shown in Figures 3 and 4. In Figure 4D the Ki constant for inhibiting FAM-ISRIB binding by eIF2aP alone appears significantly weaker than that for the full eIF2P complex shown in Figure 4C. This would suggest that eIF2 complex components in addition to eIF2alpha do have an impact on competition with ISRIB (point #10 of reviewer #2). If eIF2alpha's affinity is much weaker, it may explain differences when comparing with in vivo experiments as commented by reviewer #3.

2) The comparisons shown in Figures 1E, 3D, and 5 need to be supported by statistics (reviewer #1).

3) Using a novel in vivo fluorescence assay the authors find no evidence for complexes lacking eIF2Balpha in the cells they tested. The authors suggest that when eIF2Balpha is compromised in VWM/CACH disease, promoting complex stability may be an additional mechanism of ISRIB action. The authors should provide some direct evidence to support this, for example by testing a VWM mutation that reduces eIF2Balpha association with eIF2B complex (https://elifesciences.org/articles/32733) (reviewer #2). In the latter paper in *eLife* the eIF2Balpha-V183F mutation destabilizes the decamer and reduces eIF2B activity in extracts, but does not activate the ISR. Addition of ISRIB restores GDP release and suppresses Tg-induced ISR in the mutant cells. As this eIF2Balpha mutant had no phenotype in the absence of stress (Figure 4C,D,F), the authors are not expected to see an effect. However, their new fluorescence assay could be a more accurate reporter than the assays used in the Wong paper which rely on biochemical fractionations- which likely report destabilization upon dilution/extraction.

4) There is a considerable amount of prior excellent work on this topic (starting with the discovery of ISRIB by the Walter group, and the recent paper of Ron's group in Molecular Cell). Therefore, some further references could be included in the Introduction and further discussed (see for example, reviewer #2)

5) The timelines for the addition of dTag13, trimethoprim, ISRIB and/or Tg to cells should be described in better detail, as it is unclear in most experiments for how long trimethoprim is added prior to the stated other treatments.

*Reviewer #1 (Recommendations for the authors):*

In this paper Schoof et al. describe the allosteric communication between the physically distant eIF2, eIF2-P, and ISRIB binding sites. The paper documents a series of well-designed experiments that shed light on the mechanism of action of ISRIB, a potent inhibitor of the integrated stress response (ISR). The authors report that ISRIB inhibits the ISR using two mechanisms of action. By revealing an allosteric communication between the physically distant eIF2, eIF2-P, and ISRIB binding sites, the new findings significantly extend the findings of Zyryanova et al., 2020. In addition, the authors show that ISRIB promotes eIF2B assembly under conditions where eIF2Bα is limiting or decamer stability is compromised. Thus, ISRIB negate the ISR using two, likely not mutually exclusive, different modalities.

The following are comments which the authors should consider.

1) The comparisons shown in Figures 1E, 3D, and 5 need to be supported by statistics.

2) For the experiments described in Figure 3, n = X (see legend). Please indicate the correct n and specific for all of the experiments whether "n" represents biological and/or technical replicates.

3) Does dTag13 trigger translation of ATF4 in control cells?

4) Previous work in yeast, insect and mammalian cells has led to the conclusion that inhibition of eIF2Balpha failed to activate the ISR as determined by measuring general translation of translation of specific mRNAs (Fabian et al., JBC, 1997; Kimball et al., JBC, 1998; Pavitt et al., MCB, 1997, Elsby et al., 2011; Dever et al., Cell, 1992). Moreover, inhibition of eIF2Bα and its yeast orthologue GCN3, make eIF2B less sensitive to inhibition by eIF2-P. In contrast, Figure 1 shows that degradation of eIF2Balpha triggers the ISR. How can these apparent contradictory results be explained? A discussion is necessary.

5) The degron-mediated degradation of eIF2Bα leads to a similar phenotype to that observed with mutations in eIF2B in VWMD. Could the authors speculate the mechanism by which degradation of the α subunit of eIF2B activates the ISR?

*Reviewer #2 (Recommendations for the authors):*

1) The existence of a recent study describing an allosteric model for ISRIB action should be acknowledged much earlier in the manuscript, it only appears very late in the Discussion. It would be appropriate to discuss in the Introduction and perhaps modify some of the experimental rationale to take this into account.

2) “Both eIF2 and eIF2-P binding sites span interfaces between eIF2B subunits present in the decamer but not in the subcomplexes from which it is assembled.”

The authors are selective in their proposed eIF2B assembly pathway options which are stated as a fact. There is ample evidence that eIF2Babd hexamer and eIF2Bge dimers can assemble, yet these intermediates are not discussed or evaluated anywhere in the manuscript. An 2Babd hexamer should have both binding sites for eIF2a and 2aP and its assembly could be assessed in the authors in vitro fluorescence complex assembly assay.

3) The first section headings of the Results are misleading/overstating the results, because the assembly state of eIF2B is not monitored in the assays described. The data may be consistent with the interpretation, but they should be re-worded. The assembly state is first tested in Figure 2.

4) Figure 2. Was in vitro assembly with eIF2Bbd and 2Ba alone in the absence of eIF2Bge ever investigated in the in vitro system? Other studies (as indicated above) have indicated eIF2Bbd tetramer and 2Babd hexamers can form.

5) Can the authors separate whether ISRIB at sub-saturating concentrations primarily promotes the formation of octamers from tetramers or instead binds to and stabilises octamers that form independently and so acts to retard the rate of disassembly of octamers back into tetramers that can otherwise occur in the experiment? My reading of the data suggests the latter is more consistent with these experiments rather than the former, but that the text is written to imply that promoting assembly is the mechanism.

6) The experimental timelines for the addition of dTag13, trimethoprim, ISRIB and/or Tg to cells are not well described in the Materials and methods (in contrast to the detailed information for western blots), and is therefore currently unclear in most experiments for how long trimethoprim is added prior to the stated other treatments.

7) Does dTag13 act to degrade free eIF2Ba only, assembled decamers only or both equally well?

8) Figure 3 B-D derive from population data, but the error bars are extremely tight. How variable are the signals from individual cells? By showing box or violin plots the variation in the population would be clear. Here the analysis is simply relying on the mean. It is unclear from the figure legend how many times these experiments have been done. It should also be possible to calculate a K1/2 constant concentration for th

9) Recent experiments have identified eIF2B foci in a range of human cells termed eIF2B bodies that respond to the ISR and ISRIB (Hodgson et al., 2019, doi: 10.1091/mbc.E18-08-0538). Are similar foci detected within the cells examined here? and if so, how do these bodies impact on the data analysis performed?

10) It should also be possible to calculate a K1/2 constant association/inhibition concentrations for the relevant plots shown in Figures 3 and 4. In Figure 4D the Ki constant for inhibiting FAM-ISRIB binding by eIF2aP alone appears significantly weaker than that for the full eIF2P complex shown in Figure 4C. This would suggest that eIF2 complex components in addition to eIF2a do have an impact on competition with ISRIB, while the authors' conclusion is ness nuanced.

11) The authors previously proposed that the binding of eIF2P to eIF2B provided a steric block through a clash between the eIF2beta-γ subunits. Now here they propose a different mechanism that is more plausible and based primarily on eIF2alphaP binding to eIF2B. Lines of published evidence that did not fit their previous model have not been cited in this context. (i) the large flexibility of eIF2alphaCTD observed many years ago in NMR experiments (the Ito et al., 2004 reference) suggests the bound eIF2g will not be in one position and (2) more recent cryoEM studies of eIF2-eIF2B complexes (Adomavicius et al., 2019 reference) that shows eIF2P anchored via the α subunit can indeed flex and move eIF2g to a range of positions that do not clash.

12) “A titration of eIF2 into this reaction allowed FAM-ISRIB polarization to recover, indicating that eIF2 binds and competes off eIF2α-P, which restores FAM-ISRIB binding.”

Are the authors suggesting that eIF2 can bind to eIF2B/eIF2P complexes forming transient complexes with >2 eIF2 molecules bound before promoting release of eIF2aP?

Is this experiment done with competing eIF2 complex or eIF2a alone?

Data in Figure 4 indicate that the eIF2 complex binds with higher affinity to eIF2B than eIF2a alone. To this reviewer, this experiment in Figure 5 is simply showing that the eIF2 complex can bind with higher affinity than the isolated eIF2a to eIF2B. What happens when both eIF2 and eIF2P complexes are used and also when both eIF2a/eIF2aP subunit only combinations are used in this assay?

13) The validation report for the EM structure indicates many regions where the level of discrepancy between the map and model is very high. The Ramachandran plot shows no unfavoured geometry, which is very unusual. This suggests that the atomic model may not be reliable. Some presentation of the model in the density in relevant regions could convince readers otherwise. There are some very badly fitting chains: I, J and A are 30 to 57% in red. It is not immediately clear which chain relates to which subunit.

14) Figure 6—figure supplement 2 the apo-structure appears to be green not grey as indicated in the legend.

*Reviewer #3 (Recommendations for the authors):*

I have only a few comments on the paper:

1) Introduction section final paragraph: "grove" should be "groove"

2) Subsection “eIF2α-P is sufficient to impair ISRIB binding to eIF2B”: the authors' conclusion that phosphorylated eIF2alpha subunit is sufficient to inhibit eIF2B is interesting. This might be consistent with the ability of overexpressed eIF2alpha-S51D to stimulate the ISR in cells. It would be interesting to know if overexpression of an eIF2alpha-S51D mutant that lacks its C-terminus and thus cannot assemble into eIF2 complexes can inhibit eIF2B in mammalian cells. Alternatively, phosphorylated eIF2alpha might only inhibit in vitro but not in vivo.

It might be worthwhile to note that the situation is likely different in yeast. Obviously, we know that yeast eIF2B is different than mammalian eIF2B in that yeast eIF2B is not activated by ISRIB. However, it was also previously shown that overexpressed eIF2alpha is readily phosphorylated in yeast (see original paper from the Hinnebusch lab, Cell 1992), but this does not lead to hyper-induction of GCN4. These yeast results were interpreted to indicate that phosphorylated eIF2alpha does not inhibit eIF2B on its own.

3) Legend of Figure 6—figure supplement 2: should "light grey" be "green"?

---

## [Author Response]

Essential Revisions:The reviewers considered your work of interest and worthy of publication in eLife after revisions in response to their criticisms. The reviewers have provided many thoughtful and constructive comments. We are therefore enclosing the full reviews, but we are listing below the comments which ought to be addressed, as follows:1) It is important to calculate a K1/2 constant of association/inhibition concentrations for the plots shown in Figures 3 and 4. In Figure 4D the Ki constant for inhibiting FAM-ISRIB binding by eIF2aP alone appears significantly weaker than that for the full eIF2P complex shown in Figure 4C. This would suggest that eIF2 complex components in addition to eIF2alpha do have an impact on competition with ISRIB (point #10 of reviewer #2). If eIF2alpha's affinity is much weaker, it may explain differences when comparing with in vivo experiments as commented by reviewer #3.

We thank the reviewer for raising these points. Regarding Figures 3 and 4, we have added E/IC_50_ values as appropriate for all panels in the figure legends. Regarding the comparison between eIF2-P and eIF2α-P, we now provide a better explanation to this critical issue in the text. In particular, the eIF2α-P vs. eIF2-P data show that for the modalities of ISRIB competition or eIF2B inhibition, eIF2 β and γ subunits are dispensable. These subunits, however, do contribute additional binding interactions, and hence their presence or absence explains different IC_50_s observed in Figure 4C and 4D. Nevertheless, we show eIF2α-P ultimately achieves the same effects as eIF2-P (both in the assays that monitor ISRIB competition and GEF inactivation). To further validate this conclusion, we now performed Michaelis Menten kinetic analyses, varying the concentrations of eIF2α-P (Figure 6A-B). We added these new data to the manuscript to further illuminate how eIF2α-P inhibits eIF2B GEF activity. We fully agree that, in the context of a cell, the extra binding energy from γ/β is important for ISR activation, yet, on mechanistic grounds, it is amply clear from our data that eIF2α-P binding is sufficient to i) change eIF2B’s conformation (Figure 7), ii) inhibit the complex (Figure 5B, 6A-B) and iii) compete off ISRIB (Figure 4D).

2) The comparisons shown in Figures 1E, 3D, and 5 need to be supported by statistics (reviewer #1).

Figures 1E, 3D, and 5 now include appropriate error bars (s.e.m.) and the legends correctly denote this fact. Additionally, for Figure 5, we have updated the figure legend to include all relevant t_1/2_ values.

3) Using a novel in vivo fluorescence assay the authors find no evidence for complexes lacking eIF2Balpha in the cells they tested. The authors suggest that when eIF2Balpha is compromised in VWM/CACH disease, promoting complex stability may be an additional mechanism of ISRIB action. The authors should provide some direct evidence to support this, for example by testing a VWM mutation that reduces eIF2Balpha association with eIF2B complex (https://elifesciences.org/articles/32733) (reviewer #2). In the latter paper in eLife the eIF2Balpha-V183F mutation destabilizes the decamer and reduces eIF2B activity in extracts, but does not activate the ISR. Addition of ISRIB restores GDP release and suppresses Tg-induced ISR in the mutant cells. As this eIF2Balpha mutant had no phenotype in the absence of stress (Figure 4C,D,F), the authors are not expected to see an effect. However, their new fluorescence assay could be a more accurate reporter than the assays used in the Wong paper which rely on biochemical fractionations- which likely report destabilization upon dilution/extraction.

We thank the reviewer for these suggestions. The VWMD mutations are thoroughly investigated in the cited paper, and ISRIB’s ability to stabilize mutant eIF2B is convincingly shown. In our view, further in vitro workup would be extraneous to the presented work and unlikely to provide significant additional information to affect our interpretation. Moreover, the generation of VWMD cells containing the reporters would be a major time- and labor-intensive endeavor that we consider well beyond the scope of this work. Such investigations are certainly of interest and should be pursued but would constitute an entire new project in its own right.

4) There is a considerable amount of prior excellent work on this topic (starting with the discovery of ISRIB by the Walter group, and the recent paper of Ron's group in Molecular Cell). Therefore, some further references could be included in the Introduction and further discussed (see for example, reviewer #2)

This is indeed a rich field with a great wealth of important papers. We have reworked our Introduction and Discussion to further discuss the state of the field and better contextualize our work. We thank the reviewers for the suggested papers. Of particular note is a more thorough assessment of the eIF2-eIF2B binding interface and the previous work by many groups to understand where/how substrate (eIF2) and inhibitor (eIF2-P) bind to eIF2B. We have additionally added a supplemental figure (Figure 1—figure supplement 1) providing a structural map to better orient the reader.

5) The timelines for the addition of dTag13, trimethoprim, ISRIB and/or Tg to cells should be described in better detail, as it is unclear in most experiments for how long trimethoprim is added prior to the stated other treatments.

We thank the reviewers for catching this oversight and have appended the Materials and methods to more accurately describe the drugging regimen, as well as inserted a statement in the Results. Unless otherwise stated, in all experiments, trimethoprim was added at the same time as the stressors (thapsigargin or dTag13) and ISRIB.

Reviewer #1 (Recommendations for the authors):In this paper Schoof et al. describe the allosteric communication between the physically distant eIF2, eIF2-P, and ISRIB binding sites. The paper documents a series of well-designed experiments that shed light on the mechanism of action of ISRIB, a potent inhibitor of the integrated stress response (ISR). The authors report that ISRIB inhibits the ISR using two mechanisms of action. By revealing an allosteric communication between the physically distant eIF2, eIF2-P, and ISRIB binding sites, the new findings significantly extend the findings of Zyryanova et al., 2020. In addition, the authors show that ISRIB promotes eIF2B assembly under conditions where eIF2Bα is limiting or decamer stability is compromised. Thus, ISRIB negate the ISR using two, likely not mutually exclusive, different modalities.The following are comments which the authors should consider.1) The comparisons shown in Figures 1E, 3D, and 5 need to be supported by statistics.

Taken care of; see above (Essential revision 2)

2) For the experiments described in Figure 3, n = X (see legend). Please indicate the correct n and specific for all of the experiments whether "n" represents biological and/or technical replicates.

This omission has been corrected.

3) Does dTag13 trigger translation of ATF4 in control cells?

This is an important point that we did test. dTag13 alone does not trigger ISR activation in cells without the eIF2B-α degron fusion, and we have added a statement to this effect in the Results supported by a supplemental figure (Figure 1—figure supplement 5). This property was actually used to screen cells to uncover clones with the integrated degron.

4) Previous work in yeast, insect and mammalian cells has led to the conclusion that inhibition of eIF2Balpha failed to activate the ISR as determined by measuring general translation of translation of specific mRNAs (Fabian et al., JBC, 1997; Kimball et al., JBC, 1998; Pavitt et al., MCB, 1997, Elsby et al., 2011; Dever et al., Cell, 1992). Moreover, inhibition of eIF2Bα and its yeast orthologue GCN3, make eIF2B less sensitive to inhibition by eIF2-P. In contrast, Figure 1 shows that degradation of eIF2Balpha triggers the ISR. How can these apparent contradictory results be explained? A discussion is necessary.

We have added a paragraph to our Discussion to cover this topic.

5) The degron-mediated degradation of eIF2Bα leads to a similar phenotype to that observed with mutations in eIF2B in VWMD. Could the authors speculate the mechanism by which degradation of the α subunit of eIF2B activates the ISR?

It does indeed lead to a similar phenotype as VWMD mutations. In Figures 1-3 we demonstrate that eIF2Bα regulates eIF2B complex assembly state and that destabilizing the complex (decamers turning into tetramers after eIF2Bα degradation), induces the ISR, and ISRIB can assemble tetramers into octamers. Thus eIF2B GEF activity is intimately tied to its assembly state, and VWMD mutations that impair complex assembly state likely activate the ISR by this mechanism.

Reviewer #2 (Recommendations for the authors):1) The existence of a recent study describing an allosteric model for ISRIB action should be acknowledged much earlier in the manuscript, it only appears very late in the Discussion. It would be appropriate to discuss in the Introduction and perhaps modify some of the experimental rationale to take this into account.

While recent work from the Ron and Ito labs came to similar conclusions regarding antagonism between eIF2-P and ISRIB, we respectfully disagree with the assessment that the Introduction and in particular our experimental rationale should be modified. Our research project and experiments were conceived long before the Ito/Ron work was publicly disclosed and published. Indeed, we asked the Ron and Ito labs to coordinate publication of our independent projects, but this proposal was not accepted. Also, our paper covers a broader scope of work. Thus we feel strongly that the current discussion is entirely appropriate and truthfully reflects the history of these discoveries.

2) “Both eIF2 and eIF2-P binding sites span interfaces between eIF2B subunits present in the decamer but not in the subcomplexes from which it is assembled.”The authors are selective in their proposed eIF2B assembly pathway options which are stated as a fact. There is ample evidence that eIF2Babd hexamer and eIF2Bge dimers can assemble, yet these intermediates are not discussed or evaluated anywhere in the manuscript. An 2Babd hexamer should have both binding sites for eIF2a and 2aP and its assembly could be assessed in the authors in vitro fluorescence complex assembly assay.

We appreciate the reviewer raising this nuanced point, and we have added a section to our Discussion, in which we discuss the possibility of other eIF2B subcomplexes. While we agree that the existence of other subspecies is of potential interest, in the context of the current manuscript and human eIF2B the eIF2Bβδγε tetramer, eIF2Bα_2_ dimer, and eIF2B(αβδγε)_2_ decamer appear to be the most abundant and physiologically relevant (Wortham et al., 2014). Further study of subcomplexes will undoubtedly be pursued in the future but is outside the scope of this work.

3) The first section headings of the Results are misleading/overstating the results, because the assembly state of eIF2B is not monitored in the assays described. The data may be consistent with the interpretation, but they should be re-worded. The assembly state is first tested in Figure 2.

While it is true that the assembly state of eIF2B is first measured in Figure 2, there is ample evidence discussed in our Introduction, and other prior work, showing that eIF2Bα and ISRIB modulate eIF2B’s assembly state in vitro. This work puts these previous observations into a cellular context and the headings accurately describe the results.

4) Figure 2. Was in vitro assembly with eIF2Bbd and 2Ba alone in the absence of eIF2Bge ever investigated in the in vitro system? Other studies (as indicated above) have indicated eIF2Bbd tetramer and 2Babd hexamers can form.

We did not investigate assembly of this potential subcomplex, partially owing to our inability to purify human eIF2Bβδ, which in our hands is not a stable subcomplex.

5) Can the authors separate whether ISRIB at sub-saturating concentrations primarily promotes the formation of octamers from tetramers or instead binds to and stabilises octamers that form independently and so acts to retard the rate of disassembly of octamers back into tetramers that can otherwise occur in the experiment? My reading of the data suggests the latter is more consistent with these experiments rather than the former, but that the text is written to imply that promoting assembly is the mechanism.

Our previous work with eIF2B tetramers, and indeed the work in this paper, show that octamers without ISRIB present are an exceedingly minor, if not non-existent, species even at high concentrations of eIF2B subunits. Therefore, the logic of ISRIB stapling two tetramers into an octamer appears far more plausible.

6) The experimental timelines for the addition of dTag13, trimethoprim, ISRIB and/or Tg to cells are not well described in the Materials and methods (in contrast to the detailed information for western blots), and is therefore currently unclear in most experiments for how long trimethoprim is added prior to the stated other treatments.

We have updated the Materials and methods and Results section to make clear when drugs are added.

7) Does dTag13 act to degrade free eIF2Ba only, assembled decamers only or both equally well?

This is an interesting question and one that is tough to address experimentally — but we can intuit a reasonable answer. Based on the data shown in Figure 2E, we know that eIF2Bα dissociates quite slowly from the assembled decamer (t_1/2_ = 3 h). We also see that essentially all eIF2Bα has been degraded by 3 h (and indeed happens much faster than that). So based on these data it is fair to assume that both free eIF2Bα and decameric eIF2Bα must be degraded.

8) Figure 3 B-D derive from population data, but the error bars are extremely tight. How variable are the signals from individual cells? By showing box or violin plots the variation in the population would be clear. Here the analysis is simply relying on the mean. It is unclear from the figure legend how many times these experiments have been done. It should also be possible to calculate a K1/2 constant concentration for th

Our analysis here (and in all plots of flow data) uses the median RFU signal. And in this case it is the ratio of two medians (donor emission / acceptor emission), making the FRET data even crisper. The relationship between these two values is fairly tight (see Author response image 1). We have made sure that the replicate number (n = 3) and EC_50_ values are reported in the figure legends.

**Author response image 1. sa2fig1:** 

9) Recent experiments have identified eIF2B foci in a range of human cells termed eIF2B bodies that respond to the ISR and ISRIB (Hodgson et al., 2019, doi: 10.1091/mbc.E18-08-0538). Are similar foci detected within the cells examined here? and if so, how do these bodies impact on the data analysis performed?

This is certainly something of interest that we hope to pursue in more appropriate cells. K562 cells (a suspension cell line with rounded morphology) are poor for microscopy work.

10) It should also be possible to calculate a K1/2 constant association/inhibition concentrations for the relevant plots shown in Figures 3 and 4. In Figure 4D the Ki constant for inhibiting FAM-ISRIB binding by eIF2aP alone appears significantly weaker than that for the full eIF2P complex shown in Figure 4C. This would suggest that eIF2 complex components in addition to eIF2a do have an impact on competition with ISRIB, while the authors' conclusion is ness nuanced.

See above (Essential Revision 1)

11) The authors previously proposed that the binding of eIF2P to eIF2B provided a steric block through a clash between the eIF2beta-γ subunits. Now here they propose a different mechanism that is more plausible and based primarily on eIF2alphaP binding to eIF2B. Lines of published evidence that did not fit their previous model have not been cited in this context. (i) the large flexibility of eIF2alphaCTD observed many years ago in NMR experiments (the Ito et al., 2004 reference) suggests the bound eIF2g will not be in one position and (2) more recent cryoEM studies of eIF2-eIF2B complexes (Adomavicius et al., 2019 reference) that shows eIF2P anchored via the α subunit can indeed flex and move eIF2g to a range of positions that do not clash.

We have inserted a sentence in the Discussion to reference this flexibility.

12) “A titration of eIF2 into this reaction allowed FAM-ISRIB polarization to recover, indicating that eIF2 binds and competes off eIF2α-P, which restores FAM-ISRIB binding.”Are the authors suggesting that eIF2 can bind to eIF2B/eIF2P complexes forming transient complexes with >2 eIF2 molecules bound before promoting release of eIF2aP?Is this experiment done with competing eIF2 complex or eIF2a alone?Data in Figure 4 indicate that the eIF2 complex binds with higher affinity to eIF2B than eIF2a alone. To this reviewer, this experiment in Figure 5 is simply showing that the eIF2 complex can bind with higher affinity than the isolated eIF2a to eIF2B. What happens when both eIF2 and eIF2P complexes are used and also when both eIF2a/eIF2aP subunit only combinations are used in this assay?

This is a fairly complicated experiment that could have been explained better. The flaw in our previous description likely contributed to a misinterpretation of the data. In brief, we show that eIF2α-P can compete off FAM-ISRIB and that a titration of eIF2 restores FAM-ISRIB binding, presumably by antagonizing eIF2α-P binding. In order to further support these claims, we have now performed Michaelis Menten kinetic analyses to examine how eIF2α-P inhibits the eIF2B, and we have performed an in vitro IP experiment to directly show antagonism between eIF2 and eIF2α-P binding to eIF2B. In aggregate, these new data show that eIF2α-P converts decameric eIF2B into “conjoined tetramers” where decameric eIF2B takes on the binding affinity and enzymatic activity of a tetramer owing to a distortion of the eIF2α binding site and an overall change of eIF2B’s conformation.

13) The validation report for the EM structure indicates many regions where the level of discrepancy between the map and model is very high. The Ramachandran plot shows no unfavoured geometry, which is very unusual. This suggests that the atomic model may not be reliable. Some presentation of the model in the density in relevant regions could convince readers otherwise. There are some very badly fitting chains: I, J and A are 30 to 57% in red. It is not immediately clear which chain relates to which subunit.

We have improved the atomic model for the apo eIF2B decamer since the initial submission and notably diminished the number of atoms that were formerly outside the map density threshold. As shown in the updated validation report and revised figures, the updated model is a reliable interpretation of the EM density. We also note the models are consistent with multiple independent structures of eIF2B determined by different labs. Chains A and B correspond to the epsilon subunits; chains I and J correspond to the γ subunits. Unlike the high-resolution core of the eIF2B complex, these subunits include peripheral domains that are more flexible – an observation that is also consistent across diverse structures of eIF2B. These regions are less well resolved (see Figure 7—figure supplement 1, panel E), resulting in worse map-to-model correlations at the periphery and more uncertainty about the precise conformations of the side chains. In prior structures, we decided to "stub" ambiguous side-chains and only modeled the backbone atoms in these challenging regions. In this case, while the peripheral regions remain challenging, we decided it was both possible and warranted to retain the side-chains during refinement. Per the reviewer's request, we added a new panel (Figure 7—figure supplement 1, panel F) to illustrate the map quality for different regions (from high to low local resolution) and how well the representative segments of the model correspond with the map in these different regions.

14) Figure 6—figure supplement 2 the apo-structure appears to be green not grey as indicated in the legend.

We thank the reviewer for catching this mistake – it has been corrected

Reviewer #3 (Recommendations for the authors):I have only a few comments on the paper:1) Introduction section final paragraph: "grove" should be "groove"

Thank you for catching this – fixed.

2) Subsection “eIF2α-P is sufficient to impair ISRIB binding to eIF2B”: the authors' conclusion that phosphorylated eIF2alpha subunit is sufficient to inhibit eIF2B is interesting. This might be consistent with the ability of overexpressed eIF2alpha-S51D to stimulate the ISR in cells. It would be interesting to know if overexpression of an eIF2alpha-S51D mutant that lacks its C-terminus and thus cannot assemble into eIF2 complexes can inhibit eIF2B in mammalian cells. Alternatively, phosphorylated eIF2alpha might only inhibit in vitro but not in vivo.It might be worthwhile to note that the situation is likely different in yeast. Obviously, we know that yeast eIF2B is different than mammalian eIF2B in that yeast eIF2B is not activated by ISRIB. However, it was also previously shown that overexpressed eIF2alpha is readily phosphorylated in yeast (see original paper from the Hinnebusch lab, Cell 1992), but this does not lead to hyperinduction of GCN4. These yeast results were interpreted to indicate that phosphorylated eIF2alpha does not inhibit eIF2B on its own.

The S51D work is very interesting, but based on the data presented in this paper (https://www.ncbi.nlm.nih.gov/pmc/articles/PMC350553/) it seems impossible to tell whether ISR activation was driven by eIF2α-S51D alone or by the subunits that were incorporated into the full heterotrimer. The experiment suggested by the reviewer of S51D overexpression lacking the C-terminus is a good one and can directly address this question. We will look into investigating it further in our systems but the outcome will not add substantially to the conclusions of the present manuscript. Regarding the point about eIF2α-P inhibiting in vitro but not in vivo, in a cellular context, the amount of free eIF2α-P may never rise to a high enough concentration to activate the ISR as the full heterotrimer binds tighter and free eIF2α-P is unlikely to ever be a major species in cells. Nevertheless, we show that a conformational change in eIF2B is responsible for a drop in eIF2B’s GEF activity and ISR activation, and this conformational change is induced by eIF2-P or eIF2α-P alone.

3) Legend of Figure 6—figure supplement 2: should "light grey" be "green"?

Thank you for catching this – fixed.